# Subcellular reorganization upon phage infection reveals stepwise assembly of viral particles from membrane-associated precursors

Simon Corroyer-Dulmont[1,2,3,4,7,9], Audrey Labarde[4,9], Vojtěch Pražák[1,3,5], Lia M. Godinho [4,8], Chloé Masson[4,7], Pierre Legrand [6], Kay Grünewald [1,2,3], Paulo Tavares [4] ✉ & Emmanuelle R. J. Quemin [1,3,4] ✉

Viruses are obligate intracellular parasites and viral infections lead to massive host cell rearrangement to support the rapid generation of progeny. Host take-over and remodelling include formation of viral-induced compartments for viral genome replication and/or assembly. While viruses infecting bacteria, bacteriophages or phages, have been extensively characterized in vitro, the molecular mechanisms underlying the viral cycle inside the crowded cytoplasm remain unclear. Here, we investigate the spatial reorganization of SPP1-infected bacteria under near-native conditions by electron cryo tomography. The most prominent feature is the formation of a large viral DNA (vDNA) compartment from which ribosomes are excluded. In SPP1 infection, there is no membrane nor proteinaceous shell surrounding these compartments. Also, we identify novel key intermediates in virus assembly: open precursors of procapsid lattice are found at the cytoplasmic membrane in a process that requires expression of the portal protein. Next, DNA-free procapsids relocate inside the vDNA compartment where vDNA is packed in a stepwise manner. Finally, DNA-filled capsids segregate to the periphery of the compartment for assembly completion and storage. Collectively, we provide comprehensive mechanistic insights into the complete viral assembly pathway of SPP1 directly *in cellula* and show how specific steps are coordinated inside the reorganized bacterial cell.

Viruses are obligatory parasites that can infect bacteria, archaea, or eukaryotes, and show co-evolution with their hosts. Several viral lineages are deeply rooted with common ancestry, such as tailed bacterial viruses (bacteriophages or phages) together with herpesviruses[1]. During lytic infection, viruses massively hijack the resources of the host towards a rapid generation of progeny. This takeover promotes reorganization of the infected cell, frequently leading to the formation of compartments dedicated to viral genome

[1]Centre for Structural Systems Biology (CSSB), Hamburg, Germany. [2]University of Hamburg (UHH), Hamburg, Germany. [3]Leibniz Institute of Virology (LIV), Hamburg, Germany. [4]Université Paris-Saclay, CEA, CNRS, Institute for Integrative Biology of the Cell (I2BC), 91198 Gif-sur-Yvette, France. [5]Department of Biochemistry, University of Oxford, Oxford, UK. [6]Synchrotron SOLEIL, L'Orme des Merisiers, Saint-Aubin, France. [7]Present address: Okinawa Institute of Science and Technology, Okinawa, Japan. [8]Present address: Technophage, Investigação e Desenvolvimento em Biotecnologia S.A., Lisboa, Portugal. [9]These authors contributed equally: Simon Corroyer-Dulmont, Audrey Labarde. ✉e-mail: paulo.tavares@i2bc.paris-saclay.fr; emmanuelle.quemin@i2bc.paris-saclay.fr

replication, progeny virion assembly, and/or evasion from host defense systems[2]. Several strategies for viral-induced compartmentation are known: membrane-bound, confined by a protein cage, formed by liquid–liquid phase separation, or a result of more complex phase separations, such as polymer compaction (e.g., DNA molecules) upon macromolecular crowding[3–5].

In the case of bacteriophages, the main steps of viral multiplication were studied extensively in vitro. However, it remains largely unclear how well this represents the molecular processes actually taking place in the complex cytoplasmic environment inside infected cells. Recently, jumbo phages were shown to form a proteinaceous shell making a nucleus-like compartment, which physically separates replication and transcription of the viral genome from translation, viral particle assembly, and metabolic reactions inside the host bacterium[6]. This feature of giant jumbo phages (genome sizes above 200 kbp) distinguishes them from the vast majority of tailed viruses that have smaller genomes, most frequently in the 30-60 kbp size range[7]. Notably, infection by siphoviruses lambda[8,9] and SPP1[10–12] also leads to the formation of a viral DNA (vDNA) compartment that is localized asymmetrically within the bacterial host cell[9,11]. In the case of SPP1, which infects the Gram-positive soil bacterium *Bacillus subtilis*[13], hybrid replisomes are found at several independent locations within the vDNA compartment. Those replisomes composed of phage (gp40) and bacterial (PolC, DnaE, DnaX, DnaN, DnaG, SsbA, and DnaB) replication proteins are hypothesized to represent foci of active vDNA replication events taking place in parallel[10]. At late stages of SPP1 infection, procapsids co-localize with vDNA for packaging[10]. Interestingly, the vDNA compartment does not protect the viral genome against bacterial defense mechanisms[12]. After vDNA packaging termination, DNA-filled viral particles are stored in so-called warehouses found outside the vDNA compartment[10].

Assembly of SPP1 particles is initiated by formation of the DNA-free procapsid I, which requires three essential components: the major capsid protein gp13, the scaffolding protein gp11 that assists with gp13 polymerization, and the portal protein gp6 located at one single vertex of the icosahedral procapsid[14]. Release of gp11 from the procapsid I interior is accompanied by a major conformational change leading to expansion of the capsid lattice to the procapsid II state[15]. The portal protein gp6 forms a dodecamer with a central channel that serves as the doorway for vDNA packaging into procapsids II. Gp1, the terminase small subunit (TerS), binds specifically to the SPP1 DNA sequence *pac* that is subsequently cleaved by gp2, the terminase large subunit (TerL)[16,17]. Docking of the terminase-DNA complex to the procapsid portal vertex leads to assembly of the DNA packaging motor that translocates DNA to the procapsid interior[16,17]. DNA packaging is terminated by cleavage of the SPP1 DNA concatemer and departure of the terminase. Sequential binding of gp15 and gp16 to the capsid portal vertex leads to formation of the connector[18] that provides the interface for attachment of the SPP1 long flexible tail, built in an independent assembly pathway[19]. Assembly of SPP1 infectious viral particles follows a pathway conserved among siphoviruses. However, there is a critical lack of knowledge on intermediate assembly states and how these processes are regulated in the bacterial cell. Here, we investigate the spatial reorganization of the host bacterial cell induced upon infection by the bacteriophage SPP1 and characterize key individual steps of the viral particle assembly pathway *in cellula* under near-native conditions. Using cellular electron cryo tomography (cryoET), we could identify assembly intermediates previously undetected in in vitro studies: formation of procapsid precursors at the cellular membrane, as well as subsequent packaging of vDNA into the procapsid structure. Importantly, we also find that these viral molecular processes correlate with a specific spatial partitioning within the rearranged infected cell, either at the plasma membrane, inside the vDNA compartment, or confined to its periphery. Altogether, we provide comprehensive mechanistic insights into the complete assembly pathway from membrane-associated precursors of procapsids to DNA-filled head-and-tail capsids directly in the bacterial host cell.

## Results

### Electron cryo tomography applied to study SPP1-infected bacteria
We developed an optimized sample preparation workflow to follow SPP1 infection *in cellula* (Supplementary Fig. 1) and measure its impact on the remodeled infected bacteria under near-native conditions[20]. The well-characterized siphovirus SPP1 infects the Gram-positive soil bacterium *Bacillus subtilis*[13]. Under optimal infection conditions, SPP1 DNA is found inside bacterial cells within 3 min post infection (p.i.)[21]. Transcription and replication of the vDNA genome are initiated rapidly afterwards[22], while expression of late genes coding for morphogenetic proteins starts at 10 to 12 min p.i[21]. From 30 min p.i. onwards, infectious viral particles are formed and then released by cell lysis[21]. In order to visualize all states of virus assembly, we generated specimens of SPP1-infected *B. subtilis* cells at 15 and 25 min p.i. by plunge-freezing (see "Methods"). Then, thin lamellae (i.e., sections of ~100 nm in thickness) were micromachined under cryogenic conditions using a scanning electron microscope equipped with a focused ion beam (cryoFIB-SEM) (Supplementary Fig. 1)[23]. Each lamella (~12 μm in width) contained cross-sections of several bacteria (Supplementary Fig. 1f) infected either with wild-type (*wt*) or mutant SPP1 phages that were investigated by cryoET (Supplementary Table 1).

### SPP1 infection leads to extensive remodeling of *B. subtilis* cells and formation of a prominent vDNA compartment
*B. subtilis* cells were imaged before and after infection with SPP1 using cryoET (Fig. 1; Supplementary Fig. 2). Tomograms of uninfected control cells display a homogeneous distribution of ribosomes in the cytoplasm (Fig. 1a; Supplementary Fig. 2a; Supplementary Movie 1), which starkly contrasts with infected cells, where ribosomes are mostly excluded from a large area of the cytoplasm (Fig. 1b; Supplementary Fig. 2b; Supplementary Movie 2). There, a prominent compartment is formed that contains the vDNA (Supplementary Fig. 2b–h). Indeed, fluorescence microscopy of ribosomal proteins L1 and S2 showed that reorganization of the cytoplasmic space correlated with the exclusion of ribosomes from the phage vDNA compartment, and late in infection with the formation of adjacent warehouses where mature infectious virions are stored (Supplementary Fig. 3). Notably, these two types of viral-induced compartments are neither delineated by a lipid bilayer nor by a proteinaceous shell (Fig. 1; Supplementary Fig. 2).

### Spatial coordination of the different stages of viral particle assembly inside the host bacterial cell
Direct visualization of individual phage particles *in cellula* using cryoET overcomes a long-standing bottleneck in the field posed by the inability to image those structures by conventional electron microscopy (EM) approaches after sample preparation on chemically fixed bacteria following infection[10]. In *wt* infections, we observed that the majority of viral particles can be assigned to several intermediates of the capsid assembly process, namely procapsid I, procapsid II, DNA-filled capsid, or mature virus (Fig. 1c–f). Structures of three of those assembly states have previously been determined from purified particles by cryoEM single particle analysis[15]. Accordingly, procapsids I are spherically-shaped icosahedra, 55 nm in diameter, with the scaffolding proteins (SP, highlighted in blue in Fig. 1c) in the lumen. Procapsids II exhibit an angular icosahedral structure with a diameter of 65 nm and were frequently filled with various amounts of DNA (Fig. 1d). Finally, DNA-filled capsids with similar dimensions to procapsids II (Fig. 1e) can be found at the periphery of the vDNA compartment, together with mature virions attached to a long flexible tail (Fig. 1f; Supplementary Table 1).

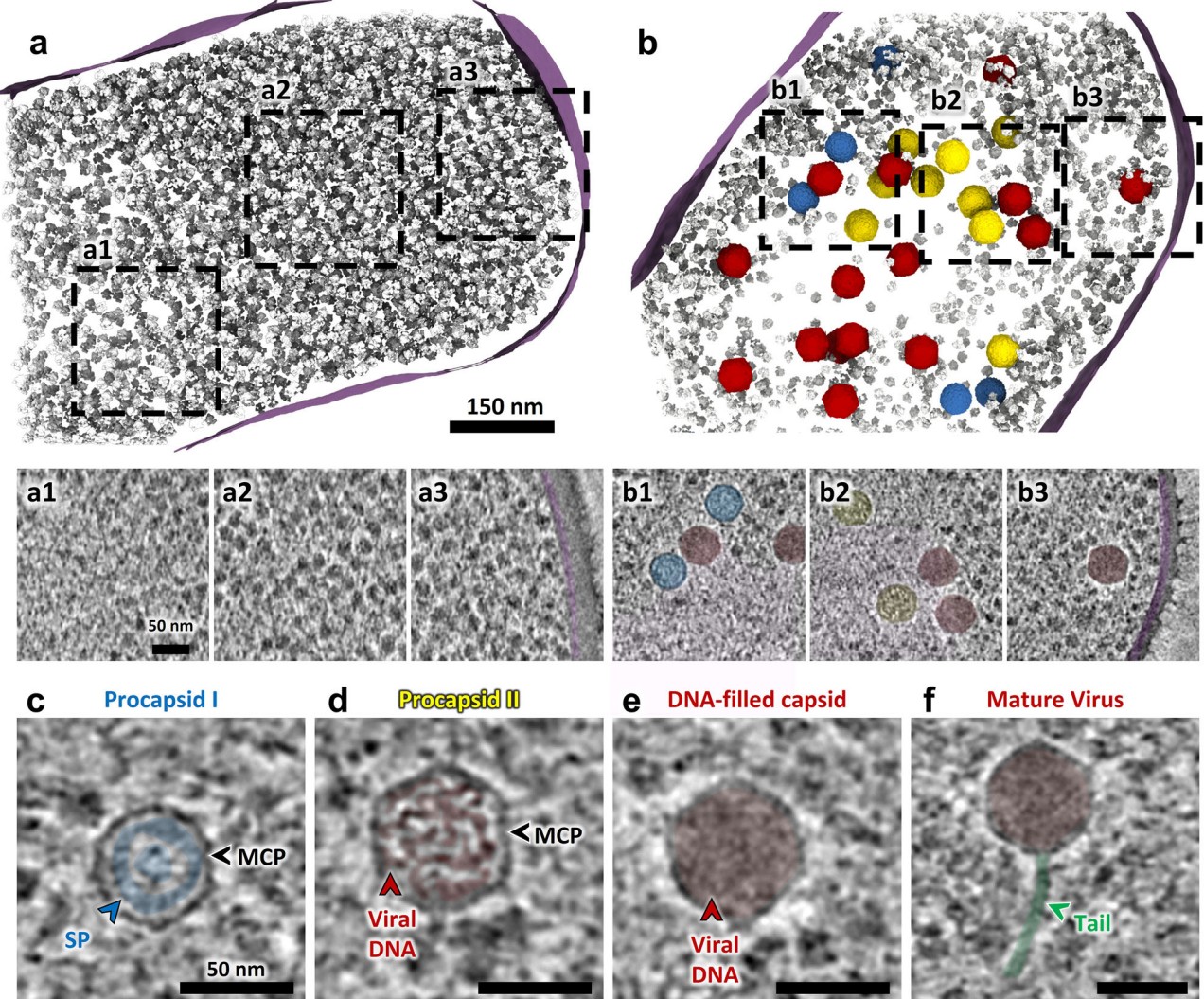

**Fig. 1 | Analysis of SPP1-induced intra-bacterial reorganization and compartmentalization of viral processes observed *in cellula*. a** Segmentation rendering and back-plotting of a representative tomogram (Supplementary Movie 1) of non-infected *B. subtilis* exhibiting the cellular organization with the bacterial cytoplasm filled with ribosomes (gray) and surrounded by the cell membrane (magenta). Zoomed-in insets of tomogram regions indicated by dashed lines in the segmentation rendering show details of a small region with bacterial DNA (**a1**) while the rest of the cytoplasm appears homogeneously filled with ribosomes (**a2, a3**). **b** Segmentation rendering and back-plotting of a representative tomogram (Supplementary Movie 2) acquired on *B. subtilis* infected by SPP1 wild type at 25 min p.i., in which a region of exclusion of ribosomes is clearly visible. Capsids at different steps of the DNA packaging process—procapsid I (blue), and procapsid II (yellow), are found in this region (detail in **b1, b2**). DNA-filled capsids (red) are present inside the vDNA compartment and at the periphery, while mature virions localize primarily at the vDNA compartment periphery (**b3**) (see distribution in Supplementary Table 1). The region of ribosome exclusion is filled with filaments that most likely represent the viral DNA (pink in **b1** and **b2**). Insets of slices from representative tomograms displaying the assembly intermediates of SPP1: procapsid I (**c**), procapsid II partially filled with DNA (**d**), DNA-filled capsid without tail (**e**), and a mature virus with a tail (**f**). Coloring of features of interest: **c** scaffolding protein (SP) in blue, major capsid protein (MCP) is indicated by black arrowheads; **d**–**f** viral DNA in red; **f** tail in green.

## Precursors of SPP1 procapsids assemble at the bacterial cell membrane

Notably, partial SPP1 procapsid I-like structures were consistently associated with the inner side of the bacterial cell membrane (Fig. 2; Supplementary Fig. 4). We consider these incomplete procapsids as early precursors of SPP1 assembly. They account for more than 10% of viral particles observed in our tomograms at 15 min p.i. and about 3% at 25 min p.i. (Supplementary Table 1). Based on the statistical distribution of precursors with different degrees of completeness, it appears that initial capsomere assembly is rapid but subsequent procapsid completion and membrane dissociation might be the limiting steps (Fig. 2i). The stages of precursor assembly observed vary from about one third of an opened sphere (Fig. 2a, e; Supplementary Fig. 4a), up to closed procapsid I-like shells still bound to the membrane (Fig. 2d, h; Supplementary Fig. 4d). The diameter of those precursors (52 nm on

average, SD = 0.7 nm, *n* = 20) as well as the curvature of the thick outer capsid lattice are similar to the unexpanded lattice of procapsids I observed in cellula (this work) and in the high-resolution structure previously determined in vitro[15] (Supplementary Fig. 5). The lumen of precursors consists of one or two layers of, presumably, internal scaffolding proteins that serve for capsomere assembly (SP, highlighted in blue in Fig. 2e–h). A pattern of radial striations is indeed observed within the outer shell, giving it the appearance of individual concentric rods (Fig. 2a–h, j). Sub-volume averaging was performed on procapsids I found in our tomograms in order to determine the symmetry of the scaffolding layers with alignment focusing either on the icosahedral shell or on the protein scaffold (see "Methods"). In both cases, no structural detail of the scaffolding proteins was resolved, regardless of whether 12-fold icosahedral symmetry was introduced or not, implying that the scaffold does not follow a defined regular

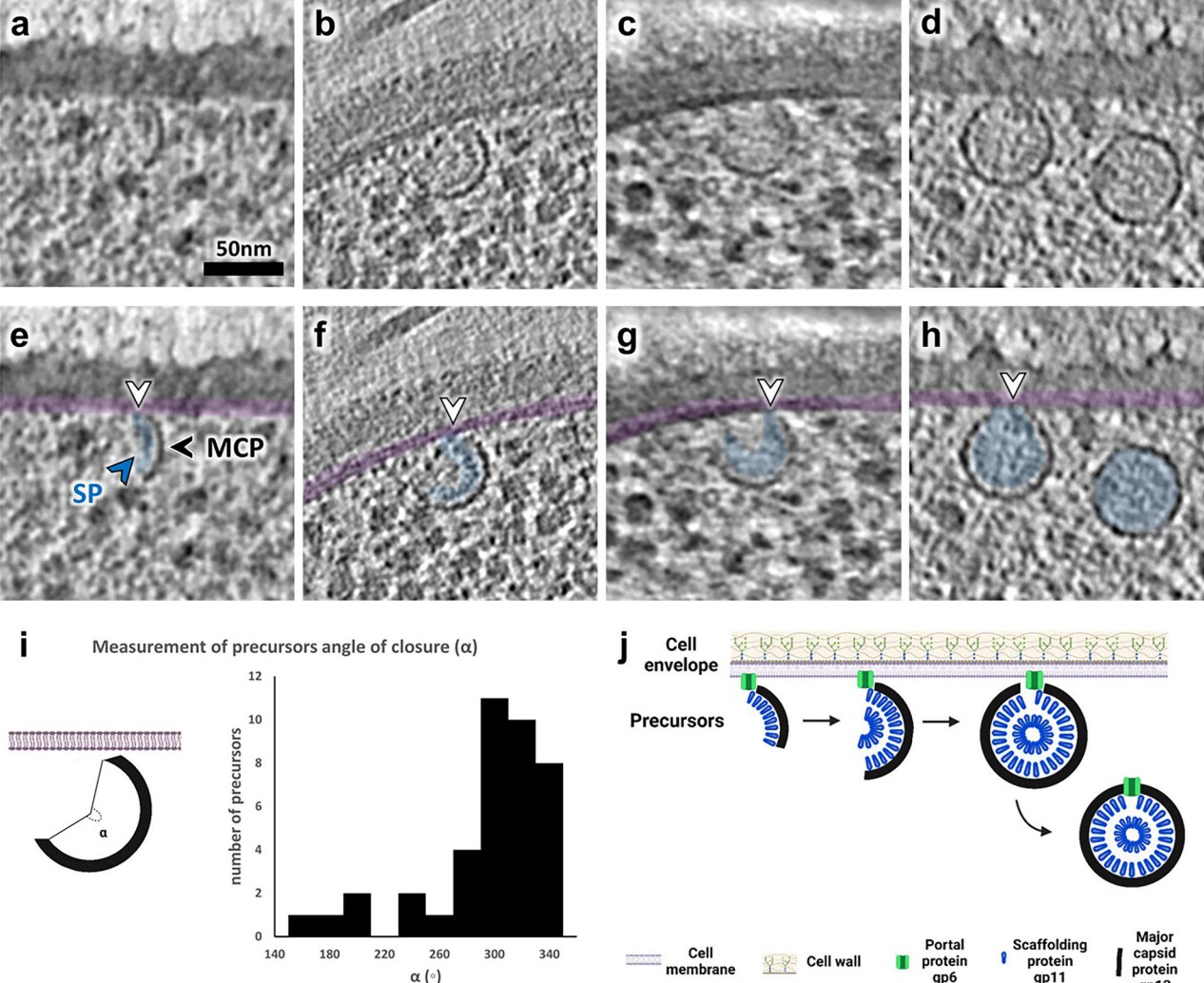

**Fig. 2 | Stepwise assembly of SPP1 procapsids is initiated by the formation of precursors at the inner side of the bacterial cell membrane. a–d** Zoomed-in insets of slices from tomograms of *B. subtilis* infected by SPP1 displaying procapsid precursors associated with the inner side of the bacterial cell membrane at different stages of procapsid lattice growth. Notably, the complete procapsid I particle is found detached from the cell membrane in (**d**). Serial slices through the tomograms are provided in Supplementary Fig. 4. **e–h** Transparent overlay coloring of slices shown in (**a–d**) to highlight the internal scaffolding protein (SP; in blue; blue arrowhead) visible in precursors and in procapsids I. The major capsid protein (MCP; black arrowhead) is also annotated, and the cell membrane is overlaid in magenta. White arrowheads indicate connections between the precursors and the cell membrane. **i** Distribution of assembly intermediates of procapsid precursors found in tomograms according to their angle of closure (α), defined as in the cartoon on the left. The cartoon representation was created in BioRender. Corroyer, S. (2026) https://BioRender.com/ dxpz8k6. Source data are provided as Source data file 1. **j** Model of assembly of procapsid precursors initiated at the membrane-attached portal protein gp6 (green), followed by curvilinear co-assembly of scaffolding (gp11, blue) and major capsid (gp13, black) proteins. Created in BioRender. Corroyer, S. (2026) https://BioRender.com/ dxpz8k6.

icosahedral organization in these procapsids but a different and possibly asymmetrical shell packing (Supplementary Fig. 5a, b).

## Role of the portal protein in membrane-associated procapsid assembly

All capsid precursors observed were in close proximity to the bacterial cell membrane, and connections are sometimes visible between the inner side of the cell membrane and the capsid lattice being assembled (Fig. 2a–h; Supplementary Fig. 4). These suggest that the host membrane serves as an anchor point for nucleation of procapsid assembly initiation. We hence hypothesized that the portal protein gp6 could nucleate SPP1 procapsid assembly at the bacterial membrane. Consistent with this hypothesis, it was already reported that gp6 is soluble in solution[24] but can also form nanopores in membranes under certain conditions in vitro[25]. In order to probe the localization of isolated SPP1 portal proteins in cellula, we engineered a *B. subtilis* strain producing

gp6 whose carboxyl terminus was fused to the fluorescent protein mCitrine (see "Methods"). Gp6-mCitrine is functional for SPP1 assembly as assessed in a trans-complementation assay (Supplementary Fig. 6). These cells were infected by a gp6 deletion mutant strain (SPP1*gp6⁻*), ensuring that the sole source of portals for incorporation in procapsids is the bacterial-encoded gp6-mCitrine. Epifluorescence microscopy observations showed a weak punctate fluorescence distribution in non-infected bacteria (Fig. 3a). We then used total internal reflection fluorescence (TIRF) microscopy to assess the dynamics of gp6-mCitrine at different stages of infection[26]. Early in infection and in non-infected bacteria, puncta of gp6-mCitrine are highly dynamic, but become immobilized in larger clusters at 23 min p.i., which coincides with viral particle assembly (Fig. 3a, b; Supplementary Movies 3 and 4). Interestingly, these fluorescent foci localized to the periphery of the vDNA compartment (white arrows in Fig. 3b). We conclude that, at this stage of infection, gp6-mCitrine proteins are mostly incorporated in

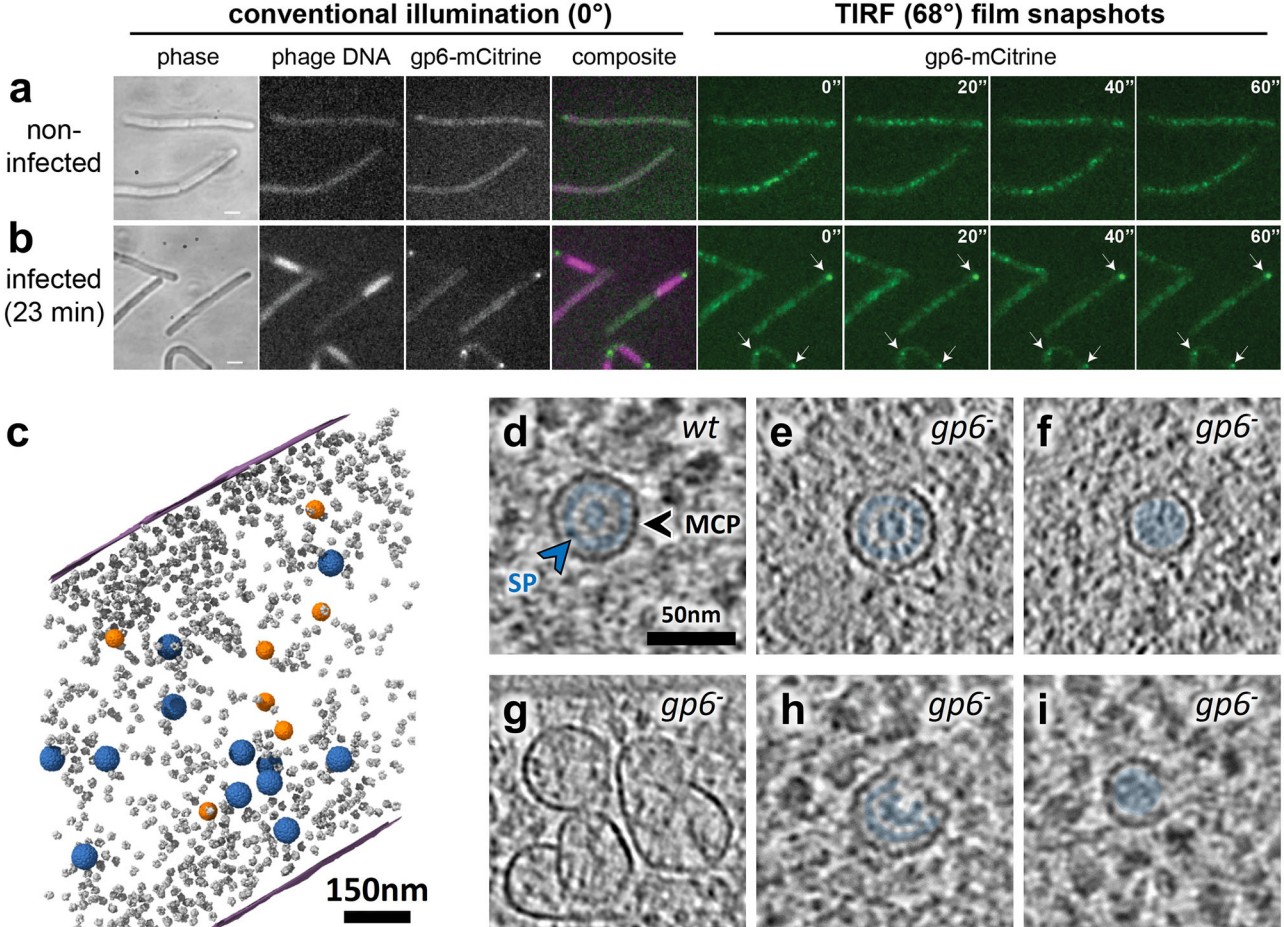

**Fig. 3 | Localization of the SPP1 portal protein inside the bacterial cell and its role in membrane-associated assembly of procapsid precursors. a** Localization of the SPP1 portal protein gp6 fused to mCitrine (gp6-mCitrine) in non-infected bacterial cells imaged by phase contrast (phase) and fluorescence. Fluorescence imaging using conventional illumination (left panels; illumination at a 0° angle) and TIRF time-lapse (right panels; illumination at a 68° angle). TIRF film snapshots show the high mobility of gp6-mCitrine foci in the peri-membrane region (right) (Supplementary Movie 3; time-lapse acquisition every 2 s during 1 min). **b** Localization of gp6-mCitrine and of phage DNA labeled by LacI-mCherry in cells infected at 23 min p.i. by SPP1*lacO64gp6⁻* that does not produce the portal protein gp6. Infected bacterial cells were imaged as in (**a**). White arrows show immobile foci of gp6-mCitrine in the TIRF time-lapse (Supplementary Movie 4). Results of experiments displayed in (**a**, **b**) are representative of three independent biological replicates. A total of 23 movies of non-infected bacteria (**a**) and 50 movies of infected bacteria (**b**) were recorded. **c** Segmentation rendering and back-plotting of a representative tomogram of *B. subtilis* infected by SPP1*gp6⁻* (Supplementary Movie 5). Procapsids I-like (blue) and smaller procapsid-like structures (~40 nm in diameter; *n* = 111; orange) are found mostly in the vDNA compartment defined by ribosome-exclusion. Ribosomes are shown in gray, and the bacterial cell membrane in magenta. **d**–**i** Zoomed-in insets of tomogram slices depicting procapsids I in *B. subtilis* infected by SPP1 wild type (*wt*) (**d**) and SPP1*gp6⁻* (*gp6⁻*) (**e**). Different structures assembled in the absence of the portal protein gp6 are also shown: small procapsid-like assembly intermediates not associated with the membrane (**f**), aberrant capsid-like material with various shapes and sizes that lack a clear inner scaffolding structure (**g**), and assembly intermediates of either procapsid I-like structure (**h**) or of a small procapsid (**i**). The scaffolding protein (SP; blue arrowhead) is annotated in (**d**) and colored (blue) when present (**d–f**, **h**, **i**). The major capsid protein (MCP; black arrowhead) forming the capsid shell is also indicated in (**d**).

viral procapsids that mature into virions and are subsequently stored in immobile warehouse compartments that were previously characterized[10].

We next investigated whether procapsid assembly occurs at the membrane in the absence of the portal protein. In *B. subtilis* cells infected with SPP1*gp6⁻*, meaning conditions under which the portal is not produced (Fig. 3c; Supplementary Fig. 2c; Supplementary Movie 5), we found that only procapsid I-like particles (Fig. 3e), smaller procapsids (Fig. 3f), and some aberrant particles (Fig. 3g) were present (Supplementary Table 1). These observations are consistent with former studies showing that SPP1 procapsid-like structures can assemble in the absence of the portal protein, but that the portal is necessary for faithful size determination of procapsid size[14]. Strikingly, no procapsid assembly intermediates were observed to be associated with the cell membrane in the 58 tomograms analyzed (Supplementary Table 1). We hence hypothesize that the incomplete,

open particles rarely spotted in the cytoplasm and in the vDNA compartment (*n* = 5) represent intermediates of an alternative, gp6-independent assembly pathway under these specific conditions (e.g., Fig. 3h, i). This previously unsuspected requirement of gp6 for localization of procapsid precursors at the inner side of the bacterial cell membrane strongly suggests that it plays a key role in regulating and driving the early stages of SPP1 procapsid assembly. Such a finely tuned mechanism would favor, in particular, the incorporation of one portal at the initiation of assembly, prior to the icosahedral lattice polymerization step.

### Concomitant DNA packaging and capsid expansion occur after relocalization of procapsids to the vDNA compartment

To determine the spatial distribution of procapsids in the absence of any vDNA packaging events, we imaged bacteria infected by SPP1 conditional mutants defective for expression of the viral terminase

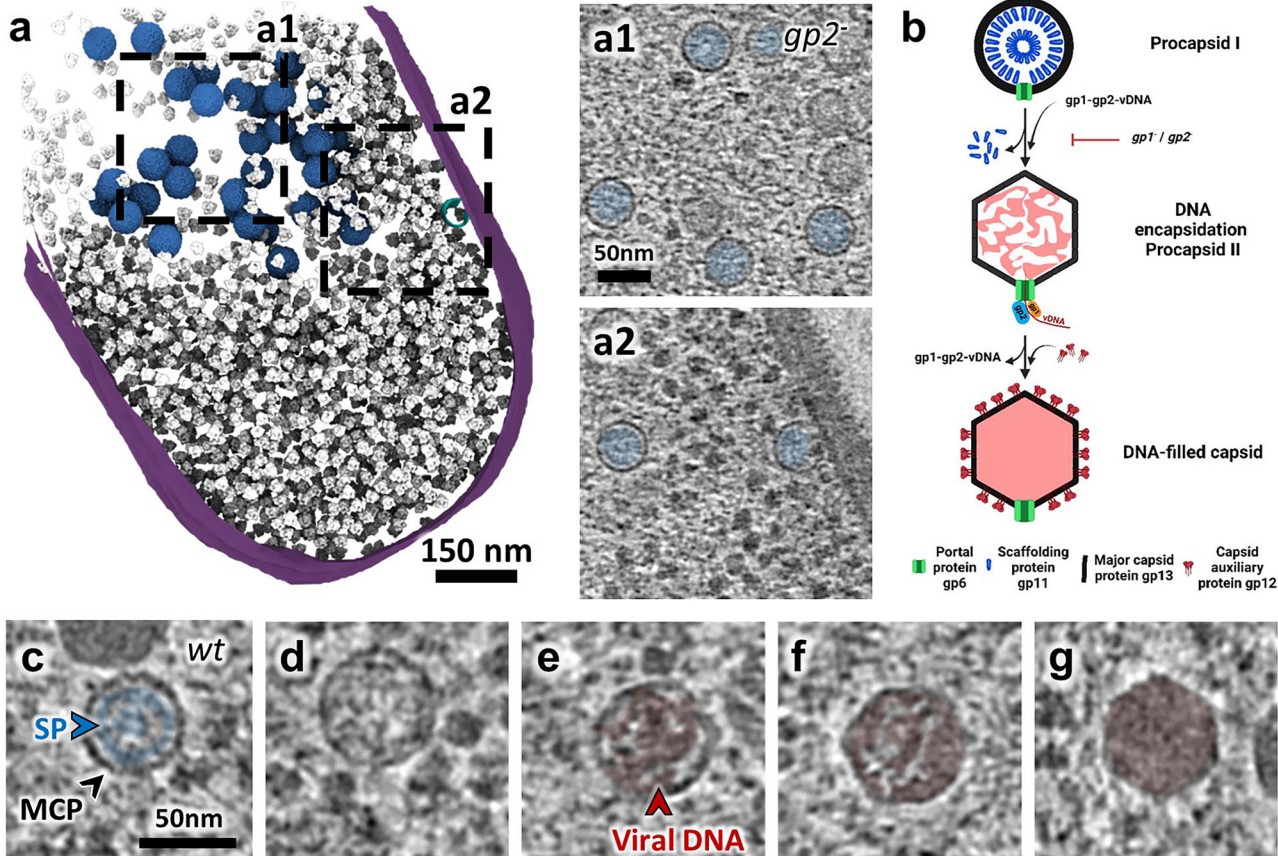

**Fig. 4 | Viral DNA packaging and capsid expansion mediate the transition from procapsid I to procapsid II in the vDNA compartment. a** Segmentation rendering and back-plotting of a representative tomogram of *B. subtilis* infected by SPP1*lacO64gp2⁻* (*gp2*) (Supplementary Movie 6), a mutant defective in viral DNA packaging. The bacterial cell membrane is shown in magenta, back-plotted ribosomes in gray, the procapsid precursor in cyan, and procapsid I in blue. Zoomed-in insets of regions indicated by dashed lines in the segmentation rendering show details of the vDNA compartment interior (**a1**) and periphery with a precursor at the cellular membrane (**a2**). **b** Schematic representation of viral DNA packaging steps during SPP1 assembly, including procapsid I, procapsid II, and DNA-filled capsid, as well as concomitant steps of capsid expansion and release of the scaffolding proteins. Created in BioRender. Corroyer, S. (2026) https://BioRender.com/ c0zoaiv. **c–g** Zoomed-in insets of slices from tomograms of *B. subtilis* infected by SPP1 wild type (*wt*) displaying intermediates of viral DNA genome packaging inside the vDNA compartment. **c** Detail of procapsid I with the arrangement of the scaffolding protein (SP; blue arrowhead), gp11 (blue) visible on the inside as radial striations. The major capsid protein (MCP; black arrowhead), gp13, is also indicated. **d–f** Details of individual procapsids II after capsid expansion, containing an increasing amount of viral DNA inside (red arrowhead and coloring) as packaging progresses (left to right). **g** Detail of a DNA-filled capsid showing homogeneous density of the internal content, most likely representing the final stage of viral genome packaging.

components TerS or TerL (gp1 and gp2 in SPP1, respectively; see "Methods"). In SPP1*gp1⁻* and SPP1*gp2⁻* infections, capsid precursors are found associated with the membrane and procapsids I in the vDNA compartment, in a manner similar to *wt* virus infection (Fig. 4a; Supplementary Fig. 2d, e; Supplementary Movie 6). Hence, detachment of fully assembled procapsids I from the membrane and their relocation to the vDNA compartment occur independently of subsequent vDNA encapsidation events (Supplementary Table 1). Notably, procapsid-like structures without portal assembled during SPP1*gp6⁻* infection are also found in the vDNA compartment (blue in Fig. 3c; Supplementary Fig. 2c), demonstrating that their specific targeting to the compartment is independent of the portal, and likely relying on the procapsid icosahedral lattice properties.

The transition from procapsid I to procapsid II is marked by capsid expansion and release of the scaffolding proteins[15]. The topography of the particle surface changes to a thinner and smoother lattice in procapsid II, while maintaining the overall electrostatics potential distribution (Supplementary Fig. 7). Procapsids II were present neither in SPP1*gp1⁻* nor in SPP1*gp2⁻* infections (Fig. 4a, b; Supplementary Fig. 2d, e; Supplementary Table 1) showing that vDNA packaging is essential to trigger expansion of the procapsid lattice *in cellula* (Fig. 4c–g).

Nevertheless, in a previous study, expanded procapsids II were found in extracts of lysed bacteria from an SPP1*gp1⁻* infection[15]. Those procapsids II did not contain scaffolding proteins associated with capsid hexamers, but the ones bound to the vertex pentamers were still present. Given our new findings, it seems that the disruption of the cellular environment by lysis was the trigger for procapsid I to procapsid II transition in the absence of vDNA packaging. This discrepancy with earlier findings highlights the limitations of relying uniquely on in vitro structural biology to investigate such complex molecular processes.

In SPP1*wt* infection, procapsids II, compared to procapsids I, display a larger, more angular shape similar to the structures determined in vitro[15]. In snapshots of vDNA packaging found in tomograms, the genome appears as thin threads inside procapsids II, which all have the same dimensions (63 nm in diameter ±1.1 nm; *n* = 37). Altogether, our findings suggest that expansion is essentially a single transition event triggered by vDNA packaging initiation. Different steps of genome encapsidation were visible from nearly empty to almost full icosahedral capsids (Fig. 4d–g). The viral genome appears to occupy the overall internal procapsid II space, adopting a more compact organization progressively, and no common or regular DNA encapsidation

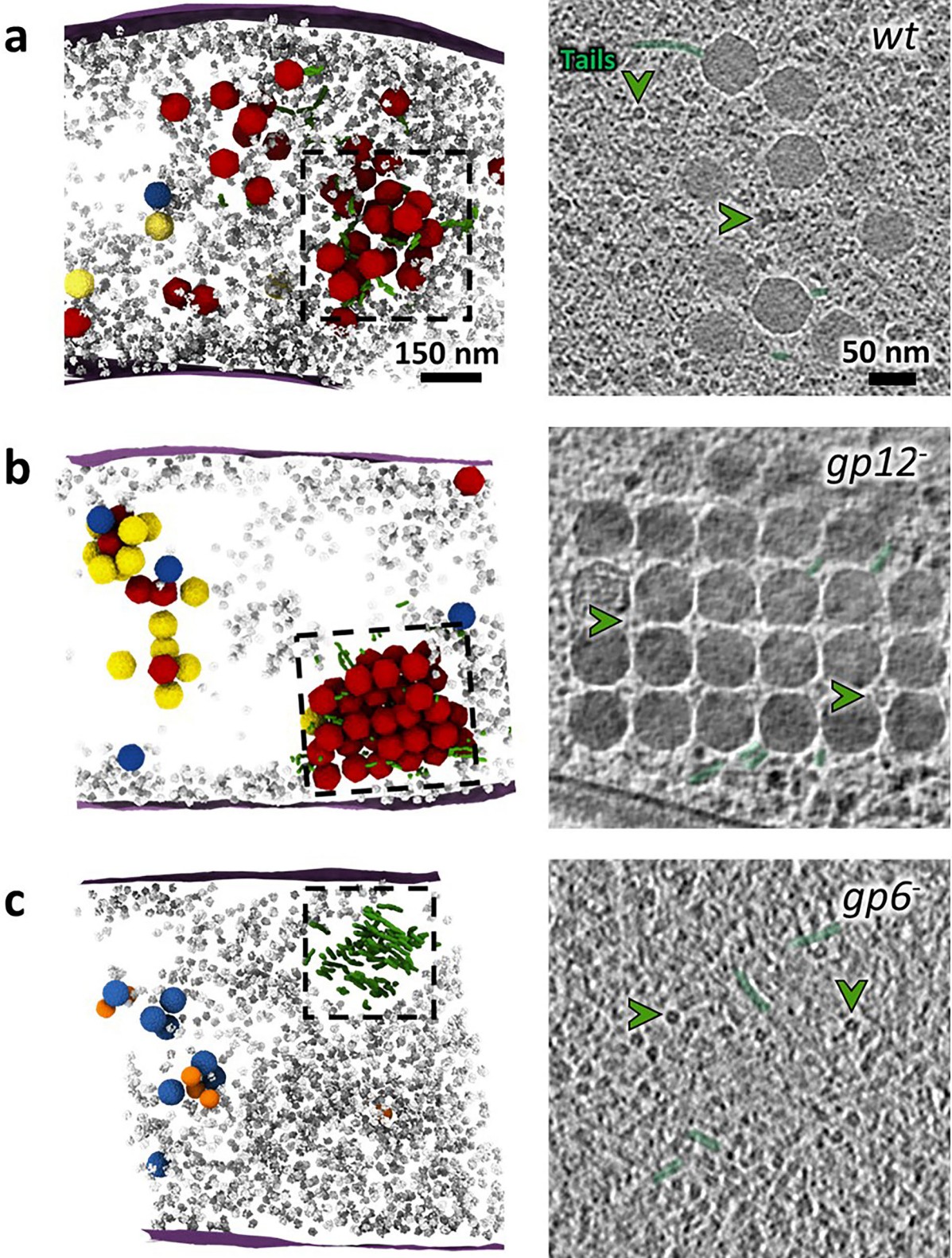

pattern was observed inside those packaging intermediates (Supplementary Fig. 8).

**Segregation of DNA-filled capsids to the periphery of the vDNA compartment and binding to pre-assembled tails**

We noticed that procapsids I and II are confined at 97% ($n = 1478$) and 94% ($n = 225$), respectively, to the vDNA compartment, consistent with the localization of vDNA encapsidation events mediating the transition between the two procapsid stages (Supplementary Table 1). In contrast, 40% of DNA-filled capsids ($n = 604$) were distinguishable by their electron-dense inner content (Figs. 1e and 4g) and the vast majority of tailed virions, 84% ($n = 113$), were located outside of this compartment, mostly at its periphery (Fig. 5a, b; Supplementary Table 1). We conclude that capsids depart from the vDNA compartment after

**Fig. 5 | Clustering of viral particles outside the vDNA compartment and binding to pre-assembled tails after viral genome packaging. a** (*Left*) Segmentation rendering and back-plotting of a representative tomogram of *B. subtilis* infected by SPP1 wild type (*wt*) with a cluster of virions at the periphery of the vDNA compartment (Supplementary Movie 7). (*Right*) Details of the cluster are indicated by dashed lines on the left panel. **b** (*Left*) Segmentation rendering and back-plotting of a representative tomogram of *B. subtilis* infected by SPP1*lacO64gp12⁻* (*gp12⁻*) showing a tightly packed warehouse at the periphery of the vDNA compartment (Supplementary Movie 8). (*Right*) Details of the warehouse are indicated by dashed lines on the left panel. **c** (*Left*) Segmentation rendering and back-plotting of a representative tomogram of *B. subtilis* infected by SPP1*gp6⁻* (*gp6⁻*), defective in assembly of functional procapsids, displaying an aggregate of pre-assembled tails (Supplementary Movie 9). (*Right*) Details of the tail aggregate are indicated by dashed lines on the left panel. Segmented elements are the bacterial cell membrane (magenta), ribosomes (gray), procapsids I (blue), procapsids II (yellow), mature virions (red), tails (green), and small procapsid-like structures (orange). Some of the tails seen in cross-sections are labeled with a green arrowhead in the right panels.

packaging of the phage genome and that tail attachment to DNA-filled capsids takes place outside of the vDNA compartment but adjacent to it (Supplementary Fig. 9a, b).

These late events of viral assembly lead to the appearance of foci detected with a fluorescently labeled marker of DNA-filled capsids[10], such as gp12-mCitrine[27]. By fluorescence microscopy, the foci are located in the vicinity of the vDNA compartment (Supplementary Fig. 3). In our tomograms, those regions appear as locally enriched in individual phage particles next to some pre-assembled tails in SPP1*wt* infections (Fig. 5a; Supplementary Fig. 2f; Supplementary Movie 7). In contrast, during infection with a SPP1*gp12⁻* mutant strain, defective in production of the capsid surface protein gp12, viral capsids are packed as large and ordered three-dimensional arrays termed warehouses, observed in 15 out of 25 tomograms (Fig. 5b; Supplementary Fig. 2g; Supplementary Movie 8). We confirm that formation of phage warehouses is a result of capsid-capsid interactions, depending on their surface properties and on the presence of the auxiliary protein gp12[27]. Indeed, particles lacking gp12 arrange as warehouses or clusters, which are significantly more compact than those containing gp12 (compare Fig. 5a, b).

Free tails are rare in infected cells when the whole assembly process of viral particles takes place, such as in *wt* infections, but can sometimes be observed at the periphery of the vDNA compartment (Fig. 5a). In contrast, clusters of tails are visible in the infected cell cytoplasm - again next to the vDNA compartment - when capsid assembly is arrested prior to tail-capsid association (Fig. 5c; Supplementary Fig. 2h; Supplementary Movie 9), as reported previously[10]. We conclude that phage tails are pre-assembled outside the vDNA compartment and bind to DNA-filled capsids after they depart from the vDNA compartment, leading to the formation of infectious particles in a spatially coordinated manner (Fig. 6).

## Discussion

Our study reports the extensive reorganization of the bacterial cytoplasmic space induced upon siphovirus SPP1 infection and its role in orchestrating progeny virion assembly. Previously, we showed that a large vDNA compartment localizes asymmetrically in the *B. subtilis* infected cell and confines viral genome replication[10]. Using an optimized sample preparation workflow for cellular cryoET (Supplementary Fig. 1; see "Methods"), we here directly visualized under near-native conditions[20] the architecture of this compartment that is delimited neither by a membrane nor by a protein cage (Fig. 1; Supplementary Fig. 2). This differs from giant jumbo phages whose vDNA is protected inside a proteinaceous lattice[6]. Nevertheless, the SPP1 vDNA compartment still excludes most of the cytoplasmic content, such as ribosomes (Fig. 1b; Supplementary Fig. 3) with their associated translation machinery, as well as certain stages of viral particle assembly (Fig. 6). The vDNA compartment is a large, irregular feature that persists throughout infection (Supplementary Fig. 2b–h). Its asymmetric position within mono-infected bacteria may be linked to the site of viral DNA ejection during entry[10,11]. Overall, this virus-induced cellular compartmentalization could arise from the folding of the large replicated SPP1 DNA concatemers into a phase-separated condensed state within the confined space of the crowded cytoplasm[4], in line with predictions from polymer dynamics simulations[5,10]. An

analogous mechanism was also proposed to drive bacterial nucleoid condensation[4]. Notably, the previous finding that replisome proteins form discrete foci associated with vDNA[10] is compatible with the hypothesis that the compartment contains sub-regions with distinct biophysical properties, including potentially more dynamic zones arising from liquid-liquid phase separation[5,28]. Further investigation will be necessary to assess such global and local properties that drive compartmentalization of vDNA and its associated transactions during infection.

Assembly of DNA-free procapsids I (Fig. 2) and tails (Fig. 5) occur outside the vDNA compartment and at different sites in the cell (Supplementary Fig. 2). Procapsids I, once assembled, relocate to the compartment (Supplementary Table 1; Supplementary Fig. 9) where initiation of DNA packaging triggers their expansion to the procapsid II conformation (Fig. 4). DNA translocation into procapsids II takes place in a stepwise manner as unraveled in our tomograms by the succession of procapsid II particles partially filled with DNA (Fig. 4d–g; Supplementary Fig. 8). The resulting DNA-filled viral particles then depart and accumulate at adjacent positions, yet spatially separated from vDNA (Fig. 1b; Supplementary Fig. 9). Finally, DNA-filled capsids bind pre-assembled tails to form viral particles that can cluster and form paracrystalline arrays called warehouses depending on the presence of the auxiliary protein gp12 (Fig. 5). Altogether, we show that the intracellular assembly pathway of SPP1 viral particles follows a precise spatially regulated program in the compartmentalized bacterium (Fig. 6).

A major finding of our study is that open procapsid-like structures are observed and associated with the inner side of the bacterial cell membrane (Fig. 2; Supplementary Fig. 4) and that such localization depends strictly on the presence of the portal protein gp6 (Fig. 3; Supplementary Table 1). These unprecedented structures likely represent initial assembly stages of DNA-free procapsids I *in cellula*. SPP1 procapsid-like structures with two different sizes and aberrant particles can assemble in the absence of the portal (Fig. 3c, e–i). However, no intermediates of their off-assembly pathway were found associated with the bacterial membrane (Fig. 3c, e–i; Supplementary Table 1). We propose a model in which the portal protein bound to the inner side of the membrane promotes association of scaffolding proteins (gp11) that then serve to initiate unidirectional polymerization of the major capsid proteins (gp13). Next, growth of the icosahedral lattice is driven by co-assembly between the scaffolding and major capsid proteins until closure of the lattice to yield procapsid I (Fig. 2). In this assembly strategy, the precise curvature of the growing lattice observed in procapsid precursors (Fig. 2i) appears to be critical for the formation of procapsids homogeneous in size. Lattice closure around the portal is proposed to promote the release of procapsid I from the membrane. Interestingly, portals and/or procapsids of a few phages were previously shown to have a membrane-associated state (see refs. 6,29–31). While the underlying molecular mechanism of this association is presently unknown, it should be the focus of future targeted research. Interestingly, the portal protein was also shown to act as the initiator of procapsid assembly in vitro for tailed phages Phi29[32] and P22[33]. Overall, our proposed model illustrates how one single specialized DNA-translocating portal vertex can be incorporated within an icosahedral lattice in a highly accurate and controlled manner (Fig. 6). Initiating assembly at the cell membrane might also ensure

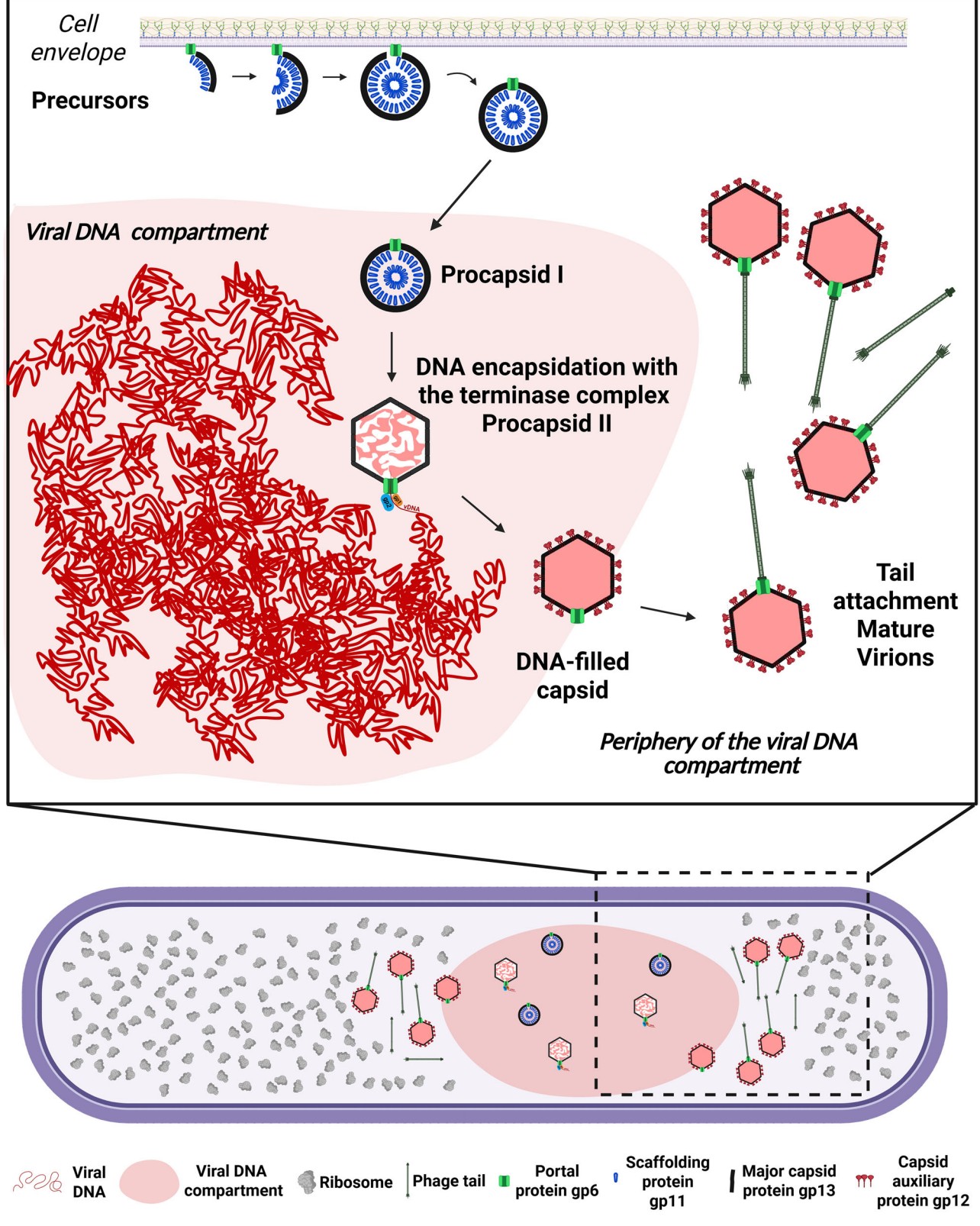

incorporation of one, and only one, portal in the procapsid and promote assembly of functional structures. Such association of phage portals with the bacterial cell membrane indeed provides a platform for procapsid assembly initiation and offers an effective way to prevent interference of additional portals with the scaffolding-major capsid polymerization reaction during the procapsid lattice growth phase in the cytoplasm.

Completion of procapsid I assembly marks its departure from the membrane to the vDNA compartment (Fig. 2d, h), where DNA packaging takes place (Fig. 4). Next, the DNA-filled capsids, bound to pre-assembled tails or not, are found outside of the compartment (Fig. 5). For each of these re-localization steps (Supplementary Fig. 9), our cryoET datasets provide no evidence for a dedicated transport system of viral particles that would be similar to the prokaryotic cytoskeleton

**Fig. 6 | SPP1-induced spatial compartmentalization of viral particle assembly in the *B. subtilis* host cell.** Model of SPP1 viral particle assembly progression inside a compartmentalized infected host bacterium. Following initiation of procapsid precursors assembly at the bacterial cell membrane, procapsids I relocate to the vDNA compartment where phage DNA replication and encapsidation take place. Initiation of viral DNA genome encapsidation triggers the structural transition from procapsid I to procapsid II that leads to the release of scaffolding proteins and capsid expansion. Subsequently, DNA-filled capsids relocate to the outside of the vDNA compartment, where they bind to pre-assembled tails. Mature virions accumulate at the periphery of the vDNA compartment and can further cluster in warehouses within the bacterial cytoplasm prior to release by host cell lysis. This schematic representation is based on our observations by cryo electron tomography of *B. subtilis* cells infected by SPP1 wild type, combined with the characterization of virus recombinant strains SPP1*gp6*⁻, SPP1lacO64*gp2*⁻, SPP1*gp1*⁻, SPP1*lacO64gp12*⁻. Note that macromolecular complexes involved in DNA translocation, such as the terminase (gp1 and gp2), are expected to associate with these structures; however, their direct visualization was not possible at the resolution of our cryoET datasets. Created in BioRender. Corroyer, S. (2026) https://BioRender.com/ pdl268e.

utilized by jumbo phages in particular[6]. With this, changes in the properties of the capsid surface at the level of the membrane-less compartment in the course of the progressing infection represent the most likely drivers for the observed relocation events. Indeed, localization of procapsids I and procapsid I-like structures in the vDNA compartment is independent of the presence of the portal protein revealing that it relies on the procapsid lattice properties (Fig. 3e, f; Supplementary Fig. 9). Individual procapsids caught in the act of genome packaging inside the vDNA compartment were found to undergo expansion at the very early stages of DNA encapsidation (Fig. 4). Subsequent to termination of genome packaging, the resulting headful DNA-filled capsids leave the vDNA compartment (Fig. 5; Supplementary Fig. 8). Although procapsid expansion dramatically alters capsid surface topology, it does not induce major changes in its overall surface electrostatic potential (Supplementary Fig. 7) that could account for the diffusive properties inside, outside or around the vDNA compartment. We therefore hypothesize that more specific, as yet unidentified, changes in the procapsid lattice, resulting from its expansion together with more subtle effects exerted by the tightly packaged vDNA on the capsid structure[15], drive the capsid expulsion from the compartment. This process appears to take place after DNA-filled capsids detach from vDNA concatemers following genome packaging termination[10]. Binding of the auxiliary protein gp12 to the DNA-filled capsid hexamers prevents formation of tightly packed warehouse compartments of viral particles (Fig. 5) similar to those previously found in thin sections of cells infected with HSV-1[34], podovirus P22[35], and polyomavirus[36]. Collectively, these findings reveal that changes in the viral capsid surface properties during its assembly pathway are the main factor driving the spatial program of viral maturation in the compartmentalized infected cell.

Bacteriophages have been used to pioneer the investigation of viral infection processes and to establish technology to probe the molecular and cellular mechanisms at play over the past 6 decades or so. This extensive work has led to major breakthroughs in our understanding of viral structure and virus-host interactions (see refs. 37–43 and references therein). In combination with sophisticated biochemistry, this research has provided detailed mechanistic and structural insights into bacteriophage assembly but lacked critical information on how they are modulated within the cellular environment. The present study expands on this extensive work and highlights the importance of the emerging field of cellular structural biology. Altogether, our new complementary findings provide a more comprehensive understanding of virus multiplication within the bacterial cell context, including in particular short-lived and transient stages of viral particle assembly inaccessible to purification for structural analysis in vitro.

## Methods
### Bacterial strains and growth conditions
Bacterial strains used in this work are listed in Supplementary Table 2.

*B. subtilis* GSY10027 was constructed by transformation of *B. subtilis* GSY10004 with plasmid pAL27 linearized with ScaI. Double cross-over led to the insertion of the gene coding for gp6-mCitrine under control of the inducible promoter $P_{xyl}$ at the *amyE locus*.

*B. subtilis* GSY10082 and GSY10087 were constructed in three steps. First, GSY10074 and GSY10085 were constructed by transformation of *B. subtilis* YB886 with plasmids pLG51 and pLG50, respectively, linearized with ScaI. Double cross-over led to the insertion of genes coding for RpsB-mCFP and RplA-mCFP, respectively, under control of the inducible promoter $P_{xyl}$ at the *amyE locus*. Then, GSY10075 and GSY10086 were constructed by transduction of GSY10074 and GSY10085, respectively, with a lysate of SPP1 wild type (*wt*) amplified in strain GSY10004. The resulting strains encode LacIΔ11-mCherry under control of the constitutive promoter $P_{pen}$ at the *thrC locus*. Finally, GSY10082 and GSY10087 were constructed by transduction of GSY10075 and GSY10086, respectively, with a lysate of SPP1*wt* amplified in strain GSY10024. The resulting strains encode gp12-mCitrine under control of the inducible promoter $P_{spac}$ at the *sacA locus*. Expression from promoters $P_{xyl}$ and $P_{spac}$ was induced with 1% (w/v) xylose and 1 mM IPTG, respectively.

Transformation of competent *B. subtilis* cells was performed by a two-step starvation protocol using Spizizen medium for growth[44]. Linearized plasmid (1 μg) was added to 500 μL competent cells for 30 min at 37 °C, plated on LB plates supplemented with 5 μg/mL kanamycin for GSY10027 or 100 μg/mL spectinomycin for GSY10074 and GSY10085, and incubated overnight at 37 °C.

Transduction of *B. subtilis* cells was performed using a previously published protocol[11]. Briefly, to produce SPP1 lysates, donor strains GSY10004 and GSY10024 were grown overnight at 30 °C in LB medium. Cultures were diluted 1:100 in 10 mL of LB medium and grown at 37 °C. At an $OD_{600nm}$ of 0.8, cultures were supplemented with 10 mM $CaCl_2$, infected with SPP1*wt* at an input multiplicity of 5 pfu/cfu (i.e., plaque-forming unit over colony-forming unit), and incubated for 2 h at 30 °C. Cell debris was sedimented by centrifugation at 4 °C for 15 min at $8000 \times g$, and the supernatant was stored at 4 °C. Then, the SPP1 transducing lysates were used to infect 300 μL of a late-logarithmic-phase culture ($OD_{600nm}$ of -1.6) of the receptor strain supplemented with 10 mM $CaCl_2$ at an input multiplicity of 1. After 10 min at 37 °C, the infected culture was mixed with 1.5 mL of pre-warmed LB medium and further incubated with shaking for 10 min. Bacteria were sedimented and resuspended in 300 μL of prewarmed LB medium containing 40 μL of anti-SPP1 serum to inactivate free phages. Bacteria were incubated for 30 min, sedimented, and resuspended in 300 mL of prewarmed LB medium. Serial dilutions of the culture were plated in solid medium supplemented with appropriate antibiotics and incubated at 37 °C overnight. Antibiotic selection was 12.5 μg/mL lincomycin + 0.5 μg/mL erythromycin (*mls* resistance) and 5 μg/mL kanamycin (*kan* resistance).

### Plasmid construction
Plasmids and oligonucleotides used in the present study are listed in Supplementary Table 2 with references.

SPP1 gene *6*, coding for the portal protein, was amplified by PCR from SPP1*lacO64* DNA with primers 55 and 56. The PCR product digested with ClaI-XhoI was ligated to pAL21 cut with the same restriction enzymes to generate plasmid pAL27.

Genes *rplA*, coding for ribosomal protein L1, and *rpsB*, coding for ribosomal protein S2, were amplified by PCR from *B. subtilis* strain

YB886 DNA with primers 20 and 28, 22 and 29, respectively. PCR products digested with KpnI-EcoRI for *rplA* or with KpnI-XhoI for *rpsB* were ligated to pAL29 cut with the same enzymes, generating pLG50 and pLG51, respectively. These constructs encode ribosomal proteins L1 and S2, respectively, fused to CFP at their carboxyl terminus. The *cfp* synthetic gene was codon-optimized for *B subtilis*. Fusion genes were expressed under the control of a xylose-inducible promoter.

Plasmid constructions were transformed into *Escherichia coli* DH5α and selected on LB plates supplemented with 100 µg/mL ampicillin. Clones were checked by DNA sequencing.

### Construction of phage strains

Phage strains used in this work are listed in Supplementary Table 2.

Phage SPP1*delX110lacO64sus115* (abbreviated SPP1*lacO64gp6⁻*) was generated by co-infecting the permissive *B. subtilis* strain HA101B with phages SPP1*delX110lacO64* (abbreviated SPP1*lacO64*) and SPP1*sus115* (abbreviated SPP1*gp6⁻*) at an input multiplicity of 10 pfu/cfu for each phage strain. Co-infection was carried out in LB medium supplemented with 10 mM CaCl₂ at 37 °C for 2 h. The cell lysate was centrifuged at 12,000 × g at 4 °C for 20 min and the supernatant was titrated using bacterial strain HA101B. Ninety-six isolated phage plaques were collected and resuspended separately in 200 µL of TBT buffer (100 mM Tris-Cl, pH 7.5, 100 mM NaCl, 10 mM MgCl₂) in a 96-well plate. Phage clones were spotted in the non-permissive strain YB886, in the permissive strain HA101B, and in the complementing strain YB886 (pSPW7) (Supplementary Table 2). Phages carrying mutation *sus115* were identified by not multiplying in strain YB886. These clones were then screened by PCR to identify those carrying insertion *lacO64* (primers 3185 and 3682; Supplementary Table 2) and deletion *delX* combined with insertion *lacO64* (primers 2677 and 3383; Supplementary Table 2)[11] Selected clones were re-titrated to obtain single plaques of pure clones, their genotype confirmed as above, and amplified in bacterial strain HA101B.

### Fluorescence microscopy

Fluorescence microscopy of GSY10082 and GSY10087 strains was performed as previously described[10]. Briefly, cultures of *B. subtilis* strains grown overnight at 30 °C were diluted 1:100 in fresh LB medium with 1 mM IPTG and 0.5% xylose and grown at 37 °C. At an OD$_{600nm}$ of 0.8, cultures were supplemented with 10 mM CaCl₂ and infected with SPP1*lacO64gp6⁻* at an input multiplicity of 1. Infected cultures were grown at 37 °C with orbital shaking for 22 min. Infected cells were mounted on agarose pads, and microscopy was carried out subsequently at room temperature. Image acquisitions were performed at early (25–35 min p.i.) and late (50–60 min p.i.) time-points after phage infection on a Zeiss Axio Observer Z1 microscope with 63X oil objective.

For Total Internal Reflection Fluorescence Microscopy (TIRFM) experiments, samples of bacterial cells were cultivated and infected with an input multiplicity of 2 pfu/cfu. After ~23 min of incubation at 37 °C with orbital shaking, cells were spotted on a 1% agarose pad (thickness of 2 mm) diluted in CH medium[45]. Initial single-exposure images were obtained with conventional epifluorescence images using 100 ms-long exposures at 488 nm and 561 nm. Then, time-lapse images were collected with TIRF using 100 ms-long exposures at 488 nm and 561 nm, with an image taken every 2 s for 1 or 2 min. Final control images illuminated by epifluorescence were collected from a single exposure of 100 ms at 488 nm. TIRFM imaging was performed using a Nikon Ti-Eclipse microscope with a 100X oil objective (NA: 1.3; WD: 0.2 mm; CFI Plan Fluor - Phase - Nikon) and an EMCCD camera (Andor). Snapshots were extracted from time-lapse movies.

### Electron cryo tomography

**Sample preparation.** *B. subtilis* strain GSY10024 was cultivated in LB medium supplemented with 1 mM IPTG at 37 °C. At OD$_{600nm}$ = 0.8 (~10$^8$ cfu/mL), the culture was supplemented with 10 mM CaCl₂, and cells were infected by SPP1 phages at 37 °C for ~15 min (early time point) or ~25 min (late time point) at an input multiplicity of 5 pfu/cfu. Cells were then centrifuged at 16,200 × g for 1 min at room temperature, and pellets were resuspended in MIII medium[46] in 1/10 of the culture's initial volume. Non-infected cells were treated the same way as a control. For electron cryo tomography (cryoET), 4 µL of infected or non-infected bacteria were added to electron microscopy (EM) grids (Quantifoil Micro Tools GmbH; Cu R2/2) before blotting for 20 s on one side and plunge freezing using a Leica EM GP plunger (Leica Biosystems Nussloch GmbH) (Supplementary Table 3).

Grids were clipped using C-clips and autoGrids in the AutoGrid assembly workstation (ThermoFisher Scientific). Lamellae were prepared in an Aquilos 2 (ThermoFisher Scientific) equipped with a gallium-focused ion beam (FIB in short), used to make thin sections of cells called lamellae by ablating cellular material around the area of interest. For this, sample preparation, milling, and polishing steps were done automatically using Maps and AutoTEM software (ThermoFisher Scientific) with a milling angle target of 8° (2° of tolerance) and a final lamella thickness set to 120 nm. The grids were retrieved after polishing of the lamellae and stored for a minimal amount of time in liquid nitrogen prior to data collection.

**Data acquisition.** Grids with milled lamellae were transferred for tilt-series acquisition to a Titan Krios G3 operated at 300 kV (ThermoFisher Scientific), equipped with a K3 direct electron detector and a post-column BioQuantum energy filter (Gatan). Grids were screened, and data acquired using SerialEM[47]. Grid scans were acquired to locate the lamellae. Tilt-series were collected using a dose-symmetric tilt scheme[48] from 60 to −60° specimen tilt range with a 3° increment step starting with 7–10° of pre-tilt (according to the specific milling angle of each lamella). Frames were acquired at 15 e⁻/pixel/s in electron counting mode with a pixel size of 3.36, 2.15, or 1.37 Å and aligned using SerialEM implementation (see Supplementary Table 3). A 70 µm objective aperture and a 20 eV energy slit were inserted during data acquisition. A nominal defocus range between 4 and 5 µm was used.

**Tomogram reconstruction.** Reconstruction of tomograms from tilt-series was done using IMOD version 4.12.40[49] or Aretomo[50] when no fiducial-like features were visible on the projection images. For visualization purposes, we typically used a binning of 4 and deconvolution using the standard parameters of ISONET without missing wedge compensation[51].

**Sub-volume averaging of procapsid I.** 572 procapsids I was manually picked using IMOD[49] from tomograms binned by 4 (*n* = 30; pixel size = 8.6 Å) and separated into two halves for independent processing done in PEET 1.14.1[52,53]. For each dataset, the initial reference was the in vitro determined map of Procapsid I (EMD-4717)[15] lowpass filtered to 55 Å. This reference was used for averaging with particles having randomized orientations using custom scripts based on TEMPy[54] with a box size of 80*80*80 voxels. The major capsid protein lattice of this average was segmented using the Segger tool[55] in Chimera[56], to obtain a mask for subsequent alignment. The whole capsid average volume using this mask converged to (stabilized on) an icosahedral structure. Alignment was then focused on the 12 icosahedral vertices using custom scripts based on TEMPy, resulting in 6864 particles extracted with a box size of 80*80*80 voxels. Further iterative alignment steps were done until convergence. The final average volume reached a resolution estimated to 26.5 Å by Fourier Shell Correlation using PEET calcUnbiasedFSC.

**Volume visualization.** Surface rendering of volumes was performed using ChimeraX. Segmentation of the bacterial cell membrane was done using drawing tools in 3dmod. Average volumes of procapsids I, procapids II, mature virions, and ribosomes were placed back within each tomogram using coordinates and orientations obtained by

sub-volume averaging with PEET 1.14.1[52,53] using custom scripts based on TEMPy[54] as described above. Here, sub-volume averaging was performed using the following references: procapsid I (EMD-4717), procapsid II (EMD-10002), mature virions (EMD-4716), and ribosomes (EMD-5787). Template matching for ribosomes was done by converting heavily filtered and thresholded volumes to model point coordinates and aligning these to a filtered reference. After cleaning by cross-correlation and removal of duplicate particles, the resulting sub-volumes were manually curated from false positives and plotted back in the original tomogram coordinate system.

## Statistical analysis
Statistical analysis was performed using RStudio software. All data is represented as a boxplot: the full line represents the value of the median, and each point corresponds to one tomogram. Shapiro and Bartlett's tests were used to evaluate normality and variance homogeneity. As the data distribution was not normal ($p$ value < 0.05), non-parametric tests were used for statistical analysis. Statistical significance for the percentage of the number of events observed for the different assembly intermediates inside the vDNA compartment was evaluated using a Kruskal-Wallis test (non-parametric one-way ANOVA test) followed by a pairwise, two-sided, Wilcoxon post-hoc test with Benjamini–Hochberg correction. Tomograms presenting aggregates of viral particles (i.e., warehouses) were removed from this Kruskal–Wallis analysis for accurate quantification of viral particles. In addition, the statistical significance of the ratio of precursors compared, this time, to the total number of capsids was evaluated using an unpaired, two-sided Wilcoxon test (non-parametric t-Student test). Statistical significance is assessed by a $p$ value < 0.05.

## Data visualization
The capsid surface electrostatic potential was calculated and displayed using PyMOL[57]. Figures were prepared with Adobe Illustrator, BioRender, Pymol, and Chimera[56].

## Reporting summary
Further information on research design is available in the Nature Portfolio Reporting Summary linked to this article.

# Data availability
The average map of procapsid I generated in this study has been deposited in the Electron Microscopy Data Bank (EMDB) under accession code EMD-54016: https://www.ebi.ac.uk/emdb/EMD-54016. Biological resources used in this work are available upon request from A.L. (audrey.labarde@i2bc.paris-saclay). Electron cryo-tomography data are available upon request from E.Q. (emmanuelle.quemin@i2bc.paris-saclay.fr). Source data are provided with this paper.

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

## Acknowledgements

The authors acknowledge funding for this collaborative project by ANR SelectVir (ANR-23-CE12-0019-01) to P.T and E.Q. In the framework of this project, S.C.D. benefited from a travel grant from the Leibniz Institute of Virology (LIV) and a Mobility fund from the Deutsch-Französische Hochschule IB-ID doctoral college to visit I2BC. E.Q. was supported by a Klaus Tschira Boost Fund in 2021 and received funding from the ATIP-Avenir program in 2022. Part of this work was performed at the CryoEM multiuser Facility at CSSB, headed by K.G. and supported by the UHH and DFG (grants INST 152/772-1, 774-1, 775-1, and 777-1). The authors also acknowledge excellent support by Carolin Seuring, Cornelia Cazey, and Ulrike Laugks, for sample preparation and data acquisition, and Wolfgang Lugmayr for supporting workflows for cryoEM/ET data processing on the Maxwell compute cluster. Grids were prepared using the Synchrotron SOLEIL facilities. The authors are indebted to Cyrille Billaudeau and Rut Carballido-Lopez (Micalis, INRAE, Jouy-en-Josas) for generously making available the Nikon epifluorescence microscope equipped with a TIRF system, for their continuous support for these experiments, and for discussions on results interpretation. The authors thank Marie-Christine Vaney (Institut Pasteur, Paris) for help with the analysis of electrostatic potentials of SPP1 capsid structures.

## Author contributions

S.C.D., A.L., K.G., P.T., and E.Q. conceived the project and obtained funding. S.C.D., A.L., P.T., and E.Q. designed the experiments. S.C.D., A.L., P.L., and E.Q. optimized sample preparation on grids. Cloning, virus production, protein expression, and complementation assay were done by A.L., L.G., and P.T. S.C.D. performed sample preparation and data collection for cryoET with supervision from E.Q. A.L. carried out the TIRF experiments and analysis. S.C.D., V.P., and E.Q. did the processing of cryoET datasets and sub-volume averaging. C.M. was responsible for the statistical analysis. S.C.D., A.L., P.T., and E.Q. prepared the manuscript with input from all co-authors.

## Funding

## Competing interests

The authors declare no competing interests.
