## [Transparent Peer Review file · Nature Communications]

Subcellular reorganization upon phage infection reveals stepwise assembly of viral particles from membrane-associated precursors

Corresponding Author: Dr Emmanuelle Queminn

Version 0:

Reviewer comments:

Reviewer #1

(Remarks to the Author)

Corroyer-Dulmont et al. report cryo-EM results of seven different SPP1-infected *Bacillus* samples (gp1⁻, gp2⁻, g12⁻, gp6⁻_15min, gp6⁻_25min, wt_15min, and wt_25min), derived from more than 200 cryo-electron tomograms. Their perseverance in collecting such a large dataset is admirable—particularly since all samples were prepared as FIB-milled lamellae to achieve sufficient contrast for downstream analysis.

Previous studies of siphovirus capsid maturation have relied primarily on *in vitro* structural analyses or *in vivo* low-resolution fluorescence microscopy. In contrast, electron cryo-tomography (ECT) provides *in situ*, high-resolution visualization. Their ECT observations are consistent with earlier *in vitro* studies (procapsid to DNA-filled capsid transition) and *in vivo* microscopy (viral DNA compartment formation). However, ECT uniquely allows direct *in situ* visualization and quantitative analysis.

Key findings from their ECT study include:

1. The membrane-less viral DNA compartment does not require a proteinaceous shell.

2. Procapsids are located within the vDNA compartment, while mature capsids are positioned at the periphery.

The most novel discovery (in my opinion) is the suggestion that gp6 initially localizes to the cell membrane, where it initiates procapsid assembly. This study represents a landmark demonstration of how ECT, when combined with optimized sample preparation, can reveal subcellular reorganization during phage assembly within the host.

While the manuscript presents valuable findings, several issues should be addressed before publication.

Major

1. Background Information – The manuscript lacks sufficient background prior to presenting results. Since phage protein nomenclature (gene products) is phage-specific, the authors should describe the overall capsid and portal architecture, including the relevant gp proteins. Prior “*in vitro*” work on the procapsid-to-mature capsid transition should also be summarized. A bit more in-depth summarization of previous *in vitro* structural and biochemical studies would help readers appreciate the novelty of the present data.

2. Virus-Induced Compartmentation – In the first two paragraphs, the authors focus on viral-induced compartmentation, emphasizing proteinaceous shells (giant jumbo phage), but provide minimal context on SPP1. Only two sentences touch upon previous SPP1 studies, without explaining what was proposed or observed. Since SPP1 vDNA compartmentation has been reported previously, more detail is required before presenting the “near-native” high-resolution cryoET results. This would make the breakthrough clearer to the field.

3. SPP1 vDNA compartment– While cryoET results demonstrate that SPP1 vDNA compartments are neither membrane- nor shell-bound, the authors should discuss potential mechanisms separating vDNA from host cytosol. Some form of validation or supporting data would strengthen the interpretation.

4. Procapsid Assembly – The gp6⁻ mutant results clearly suggest gp6 localizes to the cytosolic membrane to initiate procapsid assembly. However, portal assembly involves additional proteins, including motor proteins. What evidence supports the claim that only gp6 is required for initiation?

5. Oversimplified Figure 6 – Figure 6 illustrates capsid maturation, but the model may oversimplify the process. CryoET lacks sufficient resolution to identify all components involved, and such conclusions often require knockout mutants combined with cryoET. The discussion should acknowledge these limitations and somehow illustrate them in figure 6 to

avoid over-interpretation.

6. Unexplained Observations – Several findings require further explanation or discussion:

- o Procapsid positioning: Why are procapsids inside the vDNA compartment while mature capsids remain outside? How does DNA packaging drive this spatial transition? What surface properties change, and the evidence/reference for that?
- o gp6 dynamics: If gp6 initially binds the membrane to initiate procapsid assembly, how is it released so procapsids can enter the vDNA compartment? Are additional proteins involved? References should be cited.
- o Other assembly factors: Proteins such as terminase and motor proteins are required for DNA packaging. Where are they localized during infection? Again, those known components for capsid maturation should be included in Figure 6 and discussed in the discussion part.

Minor

1. Extended Data Fig. 6 – The scale bar (20 nm) in panel a appears inconsistent in length compared with b and c.
2. Line 306 – The phrase “Altogether, we new complementary findings provide a more comprehensive understanding of virus-host interactions” should be revised. The only virus–host interaction (in my opinion) described is gp6 binding to the host membrane.
3. References – Line 3042 mentions “phage have been studied over the past 6 decades,” yet only 31 references are cited. Additional relevant literature should be included.
4. Tomogram Data –Providing Z-stack series with segmentation would strengthen the manuscript.
5. Number of Bacteria Observed – Authors note that each tomogram contains several bacteria. It would be better to summarize the number of bacteria observed in the extended data figure 2 (this should be table, not figure)
6. Warehouse Observation– The “tightly packed warehouse” shown in gp12- strain (Fig. 5, Extended Data Fig. 3g). Was this observed only once, or repeatedly? The frequency of observation should be reported.

Reviewer #2

(Remarks to the Author)

The authors used cryo-electron tomography of *Bacillus subtilis* infected with bacteriophage SPP1 to investigate the assembly pathway of the virus in cellula. They show that the portal protein gp6 associates with the inner bacterial membrane, where it initiates nucleation of the procapsid. The assembled procapsid then relocates to the viral DNA (vDNA) compartment, where genome packaging takes place. Capsid auxiliary protein Gp12 prevents the aggregation of DNA-filled capsid. DNA-filled capsids subsequently attach to tails that are assembled separately in the cytoplasm. The observations of membrane-associated procapsid formation and genome packaging within a distinct vDNA compartment are novel and provide important insights into phage assembly in cellula. While the authors have done an excellent job visualizing the phage assembly pathway using cryo-ET, several aspects that could not be directly addressed by this technique remain unclear and would benefit from further discussion.

How does the vDNA compartment differ conceptually and functionally from the ‘replication centers’ described in other systems, given that it lacks any defined physical boundary?

A discussion of possible mechanisms leading to boundary-less compartmentalization of vDNA may help. Can it be DNA condensation or liquid-liquid phase separation-like processes, as proposed for other phage systems?

Additional discussion would be helpful to clarify whether the vDNA compartment functions to concentrate viral or host enzymes involved in DNA replication and packaging, to locally enrich nucleotides or ATP required for DNA translocation, or to protect intermediates from host defense mechanisms. It is possible that resolving this is far beyond the scope of the present paper but this should be stated.

Procapsid formation at the inner membrane: It would strengthen the discussion to include potential functional advantages of initiating procapsid assembly at the inner membrane. For example, the membrane could provide a scaffold that spatially organizes portal and capsid protein assembly, ensuring the incorporation of a single portal vertex per procapsid and preventing the formation of incomplete or aberrant structures.

While gp6 appears to play a key role in membrane-associated initiation of procapsid assembly, the molecular nature of its interaction with the membrane remains speculative. Have authors analyzed the gp6 12-mer portal structures for the protein surface that can potentially interact with the membrane?

Furthermore, the mechanism by which the portal protein detaches from the membrane after procapsid completion is not discussed. Could changes in membrane tension, capsid curvature, or scaffold release serve as potential triggers?

Minor Comments

- In Figure 1 (panels b1–b3), the vDNA compartment is highlighted in red. The same highlighting could be added to the back-plotted tomogram shown in Figure 1b for clarity. In fact, highlighting the vDNA compartment in all subsequent back-plotted tomogram figures would improve consistency and readability throughout the paper.
- The authors report using C12 symmetry for subvolume averaging of the procapsid I (lines 134–140). It should be clarified why only C12 symmetry was applied, and whether other symmetry options (e.g., C5, C6, or asymmetric reconstructions) were tested.

Reviewer #3

(Remarks to the Author)

Reviewer #4

(Remarks to the Author)

In the manuscript "Subcellular reorganization upon phage infection reveals stepwise assembly of viral particles from membrane-associated precursors," Corroyer-Dulmont et al. describe the in situ assembly of the SPP1 phage particle using cryo-electron tomography (cryoET). The excellent cryoET work is supported by well-executed fluorescence microscopy experiments and functional assays. The authors demonstrate that the SPP1 assembly pathway is consistent with those of other long-tailed phages, which were previously inferred from in vitro studies and thus carried a certain degree of ambiguity. Notably, in this manuscript, many steps of the assembly pathway have been visualized under native conditions for the first time.

The authors find that different components of the virion are assembled at spatially separated areas of the cell, which they refer to as "compartments." However, these "compartments" are not separated from the rest of the cytoplasm by either proteinaceous or lipid "walls," unlike those of jumbophages. In principle, there would be no need to bring up jumbophages if the SPP1 "compartments" were referred to by terms that do not imply a physical barrier - such as "area," "region," or "subvolume."

This is my only, and very minor, comment on an otherwise excellent manuscript. I am happy to congratulate the authors on this outstanding study!

Version 1:

Reviewer comments:

Reviewer #1

(Remarks to the Author)

Corroyer-Dulmont et al. report cryo-ET results of seven different SPP1-infected *Bacillus* samples (gp1=, gp2-, gp12-, gp6-15min, gp6-25min, WT 15min and WT 25min), based on 195 cryo-electron tomograms. Their perseverance in collecting such an extensive dataset is admirable—particularly given that all samples were prepared as FIB-milled lamellae to achieve sufficient contrast for downstream analyses.

Previous studies of siphovirus capsid maturation have relied primarily on in vitro structural analyses or in vivo low-resolution fluorescence microscopy. In contrast, electron cryo-tomography (ECT) enables in situ, high-resolution visualization. The authors' ECT observations are consistent with earlier in vitro findings (the transition from procapsid to DNA-filled capsid) and in vivo microscopy studies (viral DNA compartment formation). Importantly, ECT uniquely provides direct in situ visualization and enables quantitative analysis.

Key findings from this study include:

1. The membrane-less viral DNA compartment does not require a proteinaceous shell.
2. Procapsids are located within the vDNA compartment, whereas mature capsids are positioned at the periphery.

The most novel discovery, in my opinion, is the suggestion that gp6 initially localizes to the cell membrane, where it initiates procapsid assembly. Overall, this study represents a landmark demonstration of how ECT, when combined with optimized sample preparation, can be used to reveal subcellular reorganization during phage assembly within the host.

This revised manuscript adequately addresses all of my previous concerns. If I may offer one remaining comment: Supplementary Table 1 suggests that procapsid I can still form in the absence of gp6, whereas Figure 6 implies that gp6 is essential for procapsid I formation. This apparent discrepancy may be confusing to readers. It would be helpful if the authors could clarify this point when discussing their major findings (e.g., around lines 299 or 306).

Reviewer #2

(Remarks to the Author)

The authors have done an excellent job of responding to all of the reviewers and revising the paper accordingly.

Reviewer #3

(Remarks to the Author)

I co-reviewed this manuscript with one of the reviewers who provided the listed reports. This is part of the Nature Communications initiative to facilitate training in peer review and to provide appropriate recognition for Early Career

Researchers who co-review manuscripts.

Reviewer #4

(Remarks to the Author)

In the opinion of this scientist, the authors of the MS have adequately addressed all concerns raised by the review panel.

Point-by-point response to the reviewers' comments:

Reviewer #1 (Remarks to the Author):

Corroyer-Dulmont et al. report cryo-EM results of seven different SPP1-infected *Bacillus* samples (gp1⁻, gp2⁻, g12⁻, gp6⁻_15min, gp6⁻_25min, wt_15min, and wt_25min), derived from more than 200 cryo-electron tomograms. Their perseverance in collecting such a large dataset is admirable—particularly since all samples were prepared as FIB-milled lamellae to achieve sufficient contrast for downstream analysis.

Previous studies of siphovirus capsid maturation have relied primarily on in vitro structural analyses or in vivo low-resolution fluorescence microscopy. In contrast, electron cryo-tomography (ECT) provides in situ, high-resolution visualization. Their ECT observations are consistent with earlier in vitro studies (procapsid to DNA-filled capsid transition) and in vivo microscopy (viral DNA compartment formation). However, ECT uniquely allows direct in situ visualization and quantitative analysis.

Key findings from their ECT study include:

1. The membrane-less viral DNA compartment does not require a proteinaceous shell.
2. Procapsids are located within the vDNA compartment, while mature capsids are positioned at the periphery.

The most novel discovery (in my opinion) is the suggestion that gp6 initially localizes to the cell membrane, where it initiates procapsid assembly. This study represents a landmark demonstration of how ECT, when combined with optimized sample preparation, can reveal subcellular reorganization during phage assembly within the host.

While the manuscript presents valuable findings, several issues should be addressed before publication.

We thank the referee for the detailed revision of our work and the summary highlighting its novelty. We address below the different queries raised that helped to improve the manuscript.

Major

Q1.R1 (query 1, referee 1): Background Information – The manuscript lacks sufficient background prior to presenting results. Since phage protein nomenclature (gene products) is phage-specific, the authors should describe the overall capsid and portal architecture, including the relevant gp proteins. Prior “in vitro” work on the procapsid-to-mature capsid transition should also be summarized. A bit more in-depth summarization of previous in vitro structural and biochemical studies would help readers appreciate the novelty of the present data.

A1.R1 (answer 1, referee 1): We thank the referee for her/his useful suggestion. We agree that the background we initially provided on capsid assembly and its effectors was limited. We have now included a paragraph in the Introduction containing such information for SPP1 in more details in the revised version of the manuscript. It provides current knowledge on capsid assembly of tailed phages together with references that are relevant to the work presented in our manuscript (lines 72-88 of the revised manuscript with visible modifications).

Q2.R1: 2. Virus-Induced Compartmentation – In the first two paragraphs, the authors focus on viral-induced compartmentation, emphasizing proteinaceous shells (giant jumbo phage), but provide minimal context on SPP1. Only two sentences touch upon previous SPP1 studies, without explaining what was proposed or observed. Since SPP1 vDNA compartmentation has been

reported previously, more detail is required before presenting the “near-native” high-resolution cryoET results. This would make the breakthrough clearer to the field.

A2.R1: We thank the referee for this suggestion to highlight previous research that revealed SPP1-induced compartmentalization of the *Bacillus subtilis* cytoplasm. The initial findings that made the foundation for our work are now described in more detail in lines 61-71 of the Introduction part in the revised version of the manuscript.

Q3.R1: 3. SPP1 vDNA compartment– While cryoET results demonstrate that SPP1 vDNA compartments are neither membrane- nor shell-bound, the authors should discuss potential mechanisms separating vDNA from host cytosol. Some form of validation or supporting data would strengthen the interpretation.

A3.R1: This is indeed an important question in our opinion too. Our current hypothesis is that the large vDNA compartment is generated by folding of DNA to a phase condensed state leading to separation from the rest of the components of the cytosol. However, local regions within this compartment might have different biophysical properties. We address these issues in the Discussion of the revised text (lines 289-307): “Nevertheless, the SPP1 vDNA compartment still excludes most of the cytoplasmic content such as ribosomes (Fig. 1b; Extended DataSupplementary Fig. 34) and consequently proteinwith their associated translation reactionsmachinery, as well as certain stages of viral particle assembly (Fig. 6). The vDNA compartment is a large, and somehow irregular in shapefeature that persists throughout infection (Supplementary Fig. 2b-h). Its asymmetric position within mono-infected bacteria, there is a single vDNA compartment that persists mainly immobile throughout infection. Its asymmetric positioning in the cell is probably defined by the positionmay be linked to the site of viral DNA ejection during entry^{10,11}. Overall, This global behaviorthis virus-induced cellular compartmentalization can be explained by could arise from the folding of the large replicated SPP1 DNA concatemers to into a phase-separated condensed state within the confined space of the crowded cellular environmentcytoplasm^{4,10}, as predictedin line with predictions byfrom polymer dynamics simulations^{5,10,4,5}. SuchAn analogous mechanism was also proposed to drive bacterial nucleoid condensation (Surovetsev and Jacobs-Wagner 2018). Notably, tThe previous finding that replisome proteins form discrete foci associated withto vDNA (Labarde et al 2021) is compatible with the hypothesis that the compartment might havecontains regionsub regions with differentdistinct biophysical properties, including potentially more dynamic zones arising fromof liquid-liquid phase separation^{5,28}. Further investigation will be necessary to assess such global and local properties leading tothat drive compartmentalization of vDNA and its associated transactions during infection.”

Testing of this model will require extensive investigation of the dynamics of DNA molecules and different protein components located in the vDNA compartment using a combination of approaches like FRAP, FCS and SPT that represent future research avenues of high interest.

Q4.R1: 4. Procapsid Assembly – The gp6⁻ mutant results clearly suggest gp6 localizes to the cytosolic membrane to initiate procapsid assembly. However, portal assembly involves additional proteins, including motor proteins. What evidence supports the claim that only gp6 is required for initiation?

A4.R1: In our opinion, there are three lines of evidence, which taken together, support that the SPP1 portal protein gp6 is the initiator of procapsid assembly. First, co-production of gp6, the scaffolding protein gp11 and the major capsid protein gp13 in the heterologous bacterium *Escherichia coli* is necessary and sufficient to assemble procapsids that are competent to package

phage DNA leading to the formation of infectious phages (Ref. 14: Dröge et al 2000). This identifies the minimal set of phage proteins necessary for SPP1 procapsid assembly. Second, the SPP1 portal is necessary to regulate the size of the viral procapsid icosahedron (Ref. 14: Dröge et al 2000; Fig. 3), a feature that is determined at an early stage of procapsid lattice formation. Third, in the present work, it is shown that procapsid initiation at the membrane occurs in infection with mutants SPP1gp1⁻ and SPP1gp2⁻ that are recombinant strains targeting the SPP1 DNA packaging motor proteins (Fig. 4a; Supplementary Table 1). Finally, extensive *in vitro* work on bacteriophages phi29 and P22 made by the teams of Peter Prevelige and Carolyn Teschke also showed that the portal initiates procapsid assembly in these phages (lines 330-335 of the revised manuscript Discussion).

Q5.R1: 5. Oversimplified Figure 6 – Figure 6 illustrates capsid maturation, but the model may oversimplify the process. CryoET lacks sufficient resolution to identify all components involved, and such conclusions often require knockout mutants combined with cryoET. The discussion should acknowledge these limitations and somehow illustrate them in figure 6 to avoid over-interpretation.

A5.R1: We acknowledge the view of the reviewer. We would like to emphasize that figure 6 is only meant as a schematic representation of our proposed model, which is indeed based on cryoET analysis combined with characterization of recombinant mutant viruses. To highlight the limitations pointed out by the referee, Figure 6 was slightly improved and we added more information to the figure legend. This includes a sentence on the limited resolution to image some of the viral components including the terminase for example: "This schematic representation is based on our observations by cryo electron tomography of *B. subtilis* cells infected by SPP1 wild type combined with the characterization of virus recombinant strains SPP1gp6⁻, SPP1lacO64gp2⁻, SPP1gp1⁻, SPP1lacO64gp12⁻. Note that macromolecular complexes such as the terminase powering DNA translocation through the portal of procapsid II is certainly associated to those structures as represented in the panel (gp1 and gp2) but our cryoET imaging is limited in resolution and does not allow their direct visualization."

6. Unexplained Observations – Several findings require further explanation or discussion:

Q6.R1: o Procapsid positioning: Why are procapsids inside the vDNA compartment while mature capsids remain outside? How does DNA packaging drive this spatial transition? What surface properties change, and the evidence/reference for that?

A6.R1: Our data show that procapsids I are strongly enriched in the vDNA compartment and that such localization is independent of the presence of the portal. This localization depends hence exclusively on procapsid I properties and conceivably of its external lattice that is in contact with the surrounding DNA molecules. Analysis of the electrostatic potential of procapsid I, procapsid II and the DNA-filled capsid, did not reveal any striking difference that could explain the association/dissociation to DNA of those structures. The properties of the diffusion of such structures inside or outside the vDNA compartment remains to be determined. These problematics are now also addressed on lines 356-367 of the revised Discussion.

Q7.R1: o gp6 dynamics: If gp6 initially binds the membrane to initiate procapsid assembly, how is it released so procapsids can enter the vDNA compartment? Are additional proteins involved? References should be cited.

A7.R1: This is a great question. Presently we can only speculate. Our hypothesis is that closure of the procapsid lattice to fully embed the portal will promote release of the complete structure

from the membrane: "Lattice cClosure of the procapsid lattice around the portal is proposed to promote release of procapsid I from the membrane." (lines 330-331 of revised Discussion).

Q8.R1: o Other assembly factors: Proteins such as terminase and motor proteins are required for DNA packaging. Where are they localized during infection? Again, those known components for capsid maturation should be included in Figure 6 and discussed in the discussion part.

A8.R1: We anticipate that the viral DNA packaging proteins gp1 (TerS; specific SPP1 DNA binder) and gp2 (TerL; endonuclease and DNA packaging motor) are associated to procapsids in SPP1 DNA packaging intermediates, a hypothesis that is schematized in the revised Fig. 6. However, the resolution of our cryoET data does not allow their direct localization, a limitation that is now stressed in the figure legend (see also A5.R1).

Minor

Q9.R1: 1. Extended Data Fig. 6 – The scale bar (20 nm) in panel a appears inconsistent in length compared with b and c.

A9.R1: We thank the referee to spot this mistake, which has now been corrected.

Q10.R1: 2. Line 306 – The phrase "Altogether, we new complementary findings provide a more comprehensive understanding of virus-host interactions" should be revised. The only virus–host interaction (in my opinion) described is gp6 binding to the host membrane.

A10.R1: We agree with the referee. The sentence was revised accordingly to "...comprehensive understanding of virus multiplication within the bacterial cell context..." (lines 381-382 of revised Discussion)

Q11.R1: 3. References – Line 3042 mentions "phage have been studied over the past 6 decades," yet only 31 references are cited. Additional relevant literature should be included.

A11.R1: We focused our citations on major scientific and technological advances that contributed to the study of phage assembly and structure (lines 374-377 of revised version). Among the vast literature on the topic, we selected three additional references even though we are conscious that they cover only very incompletely the amazing discoveries made in this field of phage biology in the past 60 years.

Q12.R1: 4. Tomogram Data –Providing Z-stack series with segmentation would strengthen the manuscript.

A12.R1: We prepared segmentation of tomograms manually for the figure panels, which is a powerful and meaningful representation but remains somehow subjective. Therefore we think that representing the tomogram data as 3D rendering of the segmentation should be complemented with Z-stack series of the raw data from reconstructed tomograms (and filtered to a certain extent for display purposes), which we provide as supplementary figure panels or videos.

Q13.R1: 5. Number of Bacteria Observed – Authors note that each tomogram contains several bacteria. It would be better to summarize the number of bacteria observed in the extended data figure 2 (this should be table, not figure)

A13.R1: We agree with the reviewer and extended data figure 2 is now a table and referenced as one throughout the text. We are sorry if it was not clear from our description of the workflow that

each tomogram is taken at high magnification on a small region of a bacteria, however, it is actually the sections, the so-called FIB-milled lamellae, which contain several bacteria as written in the text: "Each lamella (~12 μm in width) contained cross-sections of several bacteria" (line 112). We hope this helps to clarify this aspect.

Q14.R1: 6. Warehouse Observation– The "tightly packed warehouse" shown in gp12- strain (Fig. 5, Extended Data Fig. 3g). Was this observed only once, or repeatedly? The frequency of observation should be reported.

A14.R1: Warehouses are found in 15 out of 25 tomograms taken on SPP1gp12⁻-infected bacteria. The fact that they are tightly packed correlate with fluorescence microscopy observations showing phage particles foci much brighter in SPP1gp12⁻ infected cells than in SPP1 wild type infections (Ref. 10: Labarde et al 2021). We have further clarified this point in the revised manuscript (line 267).

Reviewer #2 (Remarks to the Author):

The authors used cryo-electron tomography of *Bacillus subtilis* infected with bacteriophage SPP1 to investigate the assembly pathway of the virus in cellula. They show that the portal protein gp6 associates with the inner bacterial membrane, where it initiates nucleation of the procapsid. The assembled procapsid then relocates to the viral DNA (vDNA) compartment, where genome packaging takes place. Capsid auxiliary protein Gp12 prevents the aggregation of DNA-filled capsid. DNA-filled capsids subsequently attach to tails that are assembled separately in the cytoplasm. The observations of membrane-associated procapsid formation and genome packaging within a distinct vDNA compartment are novel and provide important insights into phage assembly in cellula. While the authors have done an excellent job visualizing the phage assembly pathway using cryo-ET, several aspects that could not be directly addressed by this technique remain unclear and would benefit from further discussion.

We thank the reviewer for the revision and interest, including the clear summary and suggestions to improve our manuscript.

Q1.R2: How does the vDNA compartment differ conceptually and functionally from the 'replication centers' described in other systems, given that it lacks any defined physical boundary?

A1.R2: In eukaryotes, the terminology "replication centers" has been used for intracellular compartments established inside the host infected cell that contain the viral DNA genome and the factors necessary for its replication (Schmid et al, 2014; doi: 10.1128/JVI.02046-13). In case of bacteriophage SPP1, viral DNA replication takes place inside independent foci within the vDNA compartment (Ref. 10: Labarde et al 2021; lines 61-71 of the revised manuscript). We hypothesize that these foci are conceptually and functionally the genome "replication centers". Here, both viral DNA replication and the stepwise assembly of DNA packaging events also occur in this compartment. Therefore, we decided to name the compartment according to the fact that it contains the viral DNA genome in high copy number rather than to assign it to a specific function.

Q2.R2: A discussion of possible mechanisms leading to boundary-less compartmentalization of vDNA may help. Can it be DNA condensation or liquid-liquid phase separation-like processes, as proposed for other phage systems?

A2.R2: This is an important and timely question that was also raised by referee 1. As described above in **A3.R1.**, our current hypothesis is that the large vDNA compartment is generated by

folding of DNA to a phase condensed state leading to separation from the cytosol but that local regions within the compartment might have different biophysical properties (lines 289-307 of revised Discussion). Of note, phage lambda has most likely a similar strategy to compartmentalize its vDNA genome and possibly coordinate DNA encapsidation events (Ref. 8: Trinh et al, 2020). In contrast, DNA replication of phage Phi29 relies on interactions with the host's MreB cytoskeleton to position phage DNA and the replication machinery (Munoz-Espin et al, 2009; <https://doi.org/10.1073/pnas.0906465106>). Its DNA replication process is thus constrained in the cell space to the MreB perimembrane localization. Other than that, we did not find in the literature studies for other phage systems and we are not aware of liquid-liquid phase separation-like processes during infection. We would be most thankful to the referee if he/she advises on relevant references that we missed.

Q3.R2: Additional discussion would be helpful to clarify whether the vDNA compartment functions to concentrate viral or host enzymes involved in DNA replication and packaging, to locally enrich nucleotides or ATP required for DNA translocation, or to protect intermediates from host defense mechanisms. It is possible that resolving this is far beyond the scope of the present paper but this should be stated.

A3.R2: This is a good suggestion. We showed previously that the SPP1 helicase gp40 together with host replication proteins PolC, DnaE, DnaX, DnaN, DnaG, SsbA and DnaB are massively recruited to discrete foci within the vDNA compartment (Ref. 10: Labarde et al, 2021). We added this information to the revised manuscript (Introduction; lines 61-71). Imaging of DNA packaging motor proteins is more problematic due to poor signal-to-noise ratio in cellula but we would like to tackle these aspects and address their role in future studies.

Q4.R2: Procapsid formation at the inner membrane: It would strengthen the discussion to include potential functional advantages of initiating procapsid assembly at the inner membrane. For example, the membrane could provide a scaffold that spatially organizes portal and capsid protein assembly, ensuring the incorporation of a single portal vertex per procapsid and preventing the formation of incomplete or aberrant structures.

A4.R2: We thank the reviewer for this insightful comment. Based on this suggestion, we develop further, in the revised Discussion, the mechanistic implications of assembly at the membrane associated-portal. First, that it provides a platform to initiate and direct polymerization of the scaffolding and major capsid proteins in a cooperative manner (lines 330-335). Second, what we consider most important: to ensure incorporation of one, and only one, portal in the procapsid that is a strict requirement to assemble a functional structure, meaning infectious viral particles ultimately (lines 329-337 of the revised manuscript).

Q5.R2: While gp6 appears to play a key role in membrane-associated initiation of procapsid assembly, the molecular nature of its interaction with the membrane remains speculative. Have authors analyzed the gp6 12-mer portal structures for the protein surface that can potentially interact with the membrane?

A5.R2: This is an interesting point raised by the reviewer. Indeed, quoting the referee: the nature of the interaction of gp6 with the membrane remains unknown. We have analyzed high-resolution structures available for the SPP1 portal to provide some hints on the molecular basis. The size of the portal itself make it difficult to apply molecular dynamics simulations to predict potential gp6-membrane interactions more reliably using conventional software like PMIPred (<https://pmipred.fkt.physik.tu-dortmund.de/>).

On the other end, the electrostatics potential displayed using Pymol for the structures of gp6 purified (left, 13-mer) or determined directly in the DNA-filled capsid (right, 12-mer), highlight a region on the gp6 stem that alternates polar and hydrophobic residues (figure below and new figure extended 6). This observation could explain why under certain conditions gp6 can insert into membranes (Zhou et al 2016; DOI: [10.1016/j.biomaterials.2016.08.002](https://doi.org/10.1016/j.biomaterials.2016.08.002)). Interestingly, a similar “oily” interface was reported for the stem of phage phi29 portal (Simpson et al 2000, doi: 10.1038/35047129). Although this could suggest that the gp6 stem enters the membrane, we feel that experimental data are necessary to probe and validate such hypothesis.

Q6.R2: Furthermore, the mechanism by which the portal protein detaches from the membrane after procapsid completion is not discussed. Could changes in membrane tension, capsid curvature, or scaffold release serve as potential triggers?

A6.R2: Our hypothesis is quite simple and relies on closure of the procapsid lattice around the portal that could promote release of procapsid I from the membrane (lines 330-331 of revised Discussion). This would explain that only fully assembled procapsids I are released from the membrane, which appears to happen quite quickly based on the fact that such procapsids are never observed anchored at the membrane in our extensive datasets. Of course, local membrane curvature or tension could also be a factor, which can not be assessed in our tomograms. Capsid curvature however appears less likely to play a role given that positioning of hexamers and pentamers leading to formation of a uniformly sized T=7 capsid, normally requires maintaining a specific curvature throughout capsid lattice growth. Scaffolding proteins release is also unlikely as it occurs during the transition from procapsid I to procapsid II, triggered during DNA packaging. Although, this model is generally accepted in the field, it is true that we cannot fully exclude the possibility that some scaffolding proteins are locally associated to the portal and would be specifically released when the procapsid lattice closes around the portal.

Minor Comments

Q7.R2: • In Figure 1 (panels b1–b3), the vDNA compartment is highlighted in red. The same highlighting could be added to the back-plotted tomogram shown in Figure 1b for clarity. In fact, highlighting the vDNA compartment in all subsequent back-plotted tomogram figures would improve consistency and readability throughout the paper.

A7.R2: We actually tried what the reviewer is proposing and were not convinced. We prepared segmentation of tomograms manually for the figure panels, which makes it very arbitrary to determine the contour of the vDNA compartment. This is due to the fact that it is not delimited

by a membrane nor a shell here and instead, we think that ribosome exclusion helps delineate the boundaries of the compartment in an indirect but also much more accurate manner using template matching for automatic detection and plot back in the coordinate system of the original tomogram.

Q8.R2: • The authors report using C12 symmetry for subvolume averaging of the procapsid I (lines 134–140). It should be clarified why only C12 symmetry was applied, and whether other symmetry options (e.g., C5, C6, or asymmetric reconstructions) were tested.

A8.R2: Thank you for noticing this, this is a typographical error. Icosahedral, not C12 symmetry, was introduced in an approach further outlined in image methods. This should also clarify the reasoning behind introducing this symmetry. Briefly, unsymmetrized alignment was first used to assess the structure with as little bias as possible. 12-fold icosahedral symmetry operators were introduced primarily as a means of increasing the amount particles for alignment, It is also plausible that a 1:1 stoichiometry of scaffold:capsomeres exists and it is the combination of low particle numbers and minor lattice flexibility prevented this structural relationship from emerging in the unsymmetrised case. Applying any symmetry operator comes with the real possibility of generating symmetrical features out of noise, however, this is where our use of PEET alignment is particularly useful: symmetry was not forced, but rather particles are added at symmetry-related orientations and are allowed to move independently. That there was no apparent correspondence between scaffold and capsid subunits then serves as a stronger evidence of their mismatch.

“In both cases, no structural detail of scaffolding proteins was resolved, regardless of whether 12-fold icosahedral symmetry was introduced or not” (lines 165-169).

Reviewer #3 (Remarks to the Author):

We thank the reviewer for her/his input and careful co-revision of the manuscript.

Reviewer #4 (Remarks to the Author):

In the manuscript “Subcellular reorganization upon phage infection reveals stepwise assembly of viral particles from membrane-associated precursors,” Corroyer-Dulmont et al. describe the in situ assembly of the SPP1 phage particle using cryo-electron tomography (cryoET). The excellent cryoET work is supported by well-executed fluorescence microscopy experiments and functional assays. The authors demonstrate that the SPP1 assembly pathway is consistent with those of other long-tailed phages, which were previously inferred from in vitro studies and thus carried a certain degree of ambiguity. Notably, in this manuscript, many steps of the assembly pathway have been visualized under native conditions for the first time.

Q1.R4: The authors find that different components of the virion are assembled at spatially separated areas of the cell, which they refer to as “compartments.” However, these “compartments” are not separated from the rest of the cytoplasm by either proteinaceous or lipid “walls,” unlike those of jumbophages. In principle, there would be no need to bring up

jumbophages if the SPP1 “compartments” were referred to by terms that do not imply a physical barrier - such as “area,” “region,” or “subvolume.”

A1.R4: This is an important point raised by the reviewer. We define compartment as a region in the (bacterial) cell that is enriched in some viral or cellular proteins and/or nucleic acids, here, while excluding other components. This broad definition is now relatively established in cell biology to include membrane-less regions that compartmentalize macromolecules or factors and biochemical reactions in the cell. Such compartments not delimited by a physical barrier result most frequently of phase separation that can have liquid, gel or quasi-solid biophysical properties (Azaldegui et al 2021; doi: 10.1016/j.bpj.2020.09.023; King JT, Shakya A 2021; doi: 10.1016/j.bpj.2021.01.033). We discuss in the revised version the potential biophysical mechanisms leading to formation of the vDNA focus and its inner DNA replication centers in the case of SPP1-infected *Bacillus subtilis* cells in the discussion in particular (lines 289-307).

This is my only, and very minor, comment on an otherwise excellent manuscript. I am happy to congratulate the authors on this outstanding study!

We are delighted by the referee’s enthusiasm for our work

Point-by-point response to the reviewers' queries:

Reviewer #1 (Remarks to the Author):

Corroyer-Dulmont et al. report cryo-ET results of seven different SPP1-infected *Bacillus* samples (gp1=, gp2-, gp12-, gp6- 15min, gp6- 25min, WT 15min and WT 25min), based on 195 cryo-electron tomograms. Their perseverance in collecting such an extensive dataset is admirable—particularly given that all samples were prepared as FIB-milled lamellae to achieve sufficient contrast for downstream analyses.

Previous studies of siphovirus capsid maturation have relied primarily on in vitro structural analyses or in vivo low-resolution fluorescence microscopy. In contrast, electron cryo-tomography (ECT) enables in situ, high-resolution visualization. The authors' ECT observations are consistent with earlier in vitro findings (the transition from procapsid to DNA-filled capsid) and in vivo microscopy studies (viral DNA compartment formation). Importantly, ECT uniquely provides direct in situ visualization and enables quantitative analysis.

Key findings from this study include:

1. The membrane-less viral DNA compartment does not require a proteinaceous shell.
2. Procapsids are located within the vDNA compartment, whereas mature capsids are positioned at the periphery.

The most novel discovery, in my opinion, is the suggestion that gp6 initially localizes to the cell membrane, where it initiates procapsid assembly. Overall, this study represents a landmark demonstration of how ECT, when combined with optimized sample preparation, can be used to reveal subcellular reorganization during phage assembly within the host.

We like to thank the reviewer for the detailed revision of our work and once more, for the nice summary highlighting its novelty. We address below her/his final query.

Q1.R1 (query 1, referee 1): This revised manuscript adequately addresses all of my previous concerns. If I may offer one remaining comment: Supplementary Table 1 suggests that procapsid I can still form in the absence of gp6, whereas Figure 6 implies that gp6 is essential for procapsid I formation. This apparent discrepancy may be confusing to readers. It would be helpful if the authors could clarify this point when discussing their major findings (e.g., around lines 299 or 306).

A1.R1 (answer 1, referee 1): This is indeed an important issue that justifies additional clarification in the manuscript text. Accordingly, we have now added some information in Results (lines 188-191) and Discussion (lines 306-308) of the revised manuscript quoting that "SPP1 procapsid-like structures with two different sizes and aberrant particles can assemble in absence of the portal (Fig. 3c, e-i). However, no intermediates of their off-assembly pathway were found associated to the bacterial membrane (Fig. 3c, e-j; Supplementary Table 1). "

Reviewer #2 (Remarks to the Author):

The authors have done an excellent job of responding to all of the reviewers and revising the paper accordingly.

We thank the reviewer for her/his interest in our work and the useful suggestions and comments that improved our manuscript very much throughout the revision process.

Reviewer #3 (Remarks to the Author):

We thank the reviewer for her/his input in co-reviewing the manuscript.

Reviewer #4 (Remarks to the Author):

In the opinion of this scientist, the authors of the MS have adequately addressed all concerns raised by the review panel.

We thank the reviewer for his/her great enthusiasm for our work along with the useful remarks during the review process that helped us improve the manuscript.